# Genetic predisposition to hypertension is associated with preeclampsia in European and Central Asian women

Valgerdur Steinthorsdottir et al.[#]

Preeclampsia is a serious complication of pregnancy, affecting both maternal and fetal health. In genome-wide association meta-analysis of European and Central Asian mothers, we identify sequence variants that associate with preeclampsia in the maternal genome at *ZNF831*/20q13 and *FTO*/16q12. These are previously established variants for blood pressure (BP) and the *FTO* variant has also been associated with body mass index (BMI). Further analysis of BP variants establishes that variants at *MECOM*/3q26, *FGF5*/4q21 and *SH2B3*/12q24 also associate with preeclampsia through the maternal genome. We further show that a polygenic risk score for hypertension associates with preeclampsia. However, comparison with gestational hypertension indicates that additional factors modify the risk of preeclampsia.

[#]A list of authors and their affiliations appears at the end of the paper.

Between 10 and 15% of pregnant women develop new-onset hypertension after 20 weeks gestation[1]. Hypertension accompanied by proteinuria is known as preeclampsia, which can progress into a serious disorder that contributes worldwide to the death of an estimated 50,000 women and up to 1 million babies annually, making it one of the principal causes of maternal and perinatal mortality[2]. Current models attribute preeclampsia to a combination of maternal susceptibility, often associated with an exaggerated inflammatory response to pregnancy and altered placental function with the release of stress factors that trigger widespread activation of the maternal vascular endothelium[3]. A familial predisposition to preeclampsia is well documented[4,5], and the evidence points to contributions from both maternal and fetal genomes. We recently showed that the fetal genome contains a preeclampsia susceptibility locus at FLT1[6]. However, previous attempts to identify maternal sequence variants associated with preeclampsia through genome-wide association scans (GWAS) or targeted gene-centric analysis have been hampered by small sample size[7–11].

Here we report the results of a meta-analysis of eight previously unreported GWAS with 9515 preeclamptic women and 157,719 controls from Europe and Central Asia as well as an expansion of our previous meta-analysis of preeclampsia offspring[6]. Maternal and fetal mortality attributable to hypertensive pregnancy disorders is higher in countries with developing maternity care systems, including those in Central Asia, an area that has been largely neglected in genome-wide studies. We present whole-genome sequences, and GWAS of maternal and fetal genomes, in data sets from Uzbekistan and Kazakhstan, and find no evidence that the genetic architecture of preeclampsia differs between the populations from Central Asia and those in Europe.

There is compelling epidemiological evidence that women with hypertension during pregnancy, including preeclampsia, are at increased risk of essential hypertension, coronary artery disease (CAD), chronic renal disease, and type 2 diabetes (T2D) later in life[12]. We have therefore explored our data using polygenic score analysis of the correlation between preeclampsia and variants associated with related phenotypes, including high BP, T2D, CAD, and BMI.

In our discovery analysis, we find five variants associating with preeclampsia through the maternal genome. All have previously been associated with BP. We further show that genetic predisposition to hypertension is a major risk factor for preeclampsia. However, comparison with gestational hypertension implies that additional factors are involved in the risk of preeclampsia.

## Results

### Association of fetal sequence variants with preeclampsia.
We evaluated the association of 12,130,433 sequence variants in individuals who were born of preeclamptic pregnancies (offspring) in our combined meta-analysis of 4630 European cases and 373,345 controls and 2145 Central Asian cases and 2027 controls (Fig. 1a and Supplementary Fig. 1). This corresponds to an effective sample size of ~9323 cases and the same number of controls in the combined analysis (Supplementary Fig. 2; see "Methods" section). To facilitate the use of Central Asian data, we sequenced whole genomes of 200 individuals from Kazakhstan and Uzbekistan generating reference data to improve imputation (Supplementary Table 1 and Supplementary Note 1). The association at the reported FLT1 locus on 13q12[6] was strengthened in this analysis, with a P-value of $3.0 \times 10^{-11}$ and OR = 1.17 (95% CI: 1.12–1.23) for the sentinel variant rs4769612 (Supplementary Table 2). This variant is in linkage disequilibrium (LD) with the previously reported rs4769613 ($r^2 = 0.99$). No other loci were associated at genome-wide significance (defined here as $P = 4 \times 10^{-9}$ after

adjusting for the number of variants tested) in this data set (Fig. 1a). In addition to the FLT1 locus, we followed up three variants with $P < 1 \times 10^{-6}$ in the meta-analysis in additional Kazakh samples ($N = 452$ cases, 361 controls). None of them reached genome-wide significance in the combined analysis (Supplementary Table 2). Follow-up genotyping in Finnish and Kazakh data sets further validated the association of rs4769612 yielding a combined P-value $4.3 \times 10^{-14}$ (Supplementary Table 2 and Supplementary Fig. 3). The conditional analysis identified a second independent variant rs9508092 at the FLT1 locus ($P = 3.0 \times 10^{-9}$ after conditioning on rs4769612) and confirmed another previously reported[6] independent signal at this locus rs71433277 ($r^2 = 0.97$ with previously reported rs12050029) ($P = 6.6 \times 10^{-6}$ after conditioning on rs4769612 and rs9508092) (Supplementary Tables 3 and 4; Supplementary Fig. 4). Functional annotation of these three independent signals shows that they all overlap regulatory regions in placental tissue of which one of the potential target genes is FLT1 (Supplementary Data 1).

### Association of maternal sequence variants with preeclampsia.
We evaluated the association of 11,796,347 sequence variants with preeclampsia in our combined meta-analysis of 7219 European maternal cases and 155,660 controls and 2296 Central Asian cases and 2059 controls (Fig. 1b and Supplementary Fig. 1). This corresponds to an effective sample size of ~12,392 cases and an equal number of controls in the combined analysis (Supplementary Fig. 2 and see "Methods" section). The controls comprised women with healthy pregnancies, with the exception of the two largest sample sets that used unselected (GOPEC) or female-only (deCODE) population controls (Supplementary Table 1 and see "Methods" section). One locus, on 16q12, reached genome-wide significance ($4 \times 10^{-9}$) in the maternal meta-analysis (Fig. 1b). We followed up the sentinel variants at this locus and 13 additional loci with $P < 1 \times 10^{-6}$ in additional samples of European and Kazakh origin ($N = 2635$ cases and 6379 controls) (Supplementary Data 2). Combining the discovery and follow-up data, rs1421085 in FTO (alpha-ketoglutarate-dependent dioxygenase) on 16q12 remained significant ($P = 1.2 \times 10^{-9}$, OR = 1.11, 95% CI: 1.07–1.1.15) (Table 1 and Supplementary Data 2). A second variant, rs259983 near ZNF831 (zinc finger protein 831) on 20q13 also reached genome-wide significance ($P = 2.9 \times 10^{-10}$, OR = 1.17, 95% CI: 1.11–1.23) (Table 1, Supplementary Data 2 and Supplementary Fig. 4). We found no evidence for heterogeneity in estimated effect size between the European and Central Asian data for any of the follow-up variants ($P_{\mathrm{het}} > 0.05$, Supplementary Data 2). Results for individual data sets are shown in Supplementary Fig. 3.

The two variants we identified as having a genome-wide significant association with preeclampsia through the maternal genome have both been reported to associate with BP in large meta-analyses[13,14]. Furthermore, the risk allele at the FTO variant rs1421085 associates with increased BMI, obesity, and a number of other traits including T2D[15].

### Preeclampsia and BP variants.
In addition to the FTO and ZNF831 variants being in known BP loci, three more variants at BP loci (MECOM/3q26, FGF5/4q21, and SH2B3/12q24) were among the variants we followed up but that did not reach genome-wide significance in the combined meta-analysis ($P = 1.2$ to $1.7 \times 10^{-8}$) (Supplementary Data 2). Therefore, we evaluated the preeclampsia association in our meta-analysis for each of the 896 established BP variants, as reported by Evangelou et al.[16]. The most significant association was found for rs16998073, upstream of FGF5, $P = 8.8 \times 10^{-8}$, followed by rs6015450 in ZNF831, $P = 4.8 \times 10^{-7}$, rs3184504, a missense variant in SH2B3,

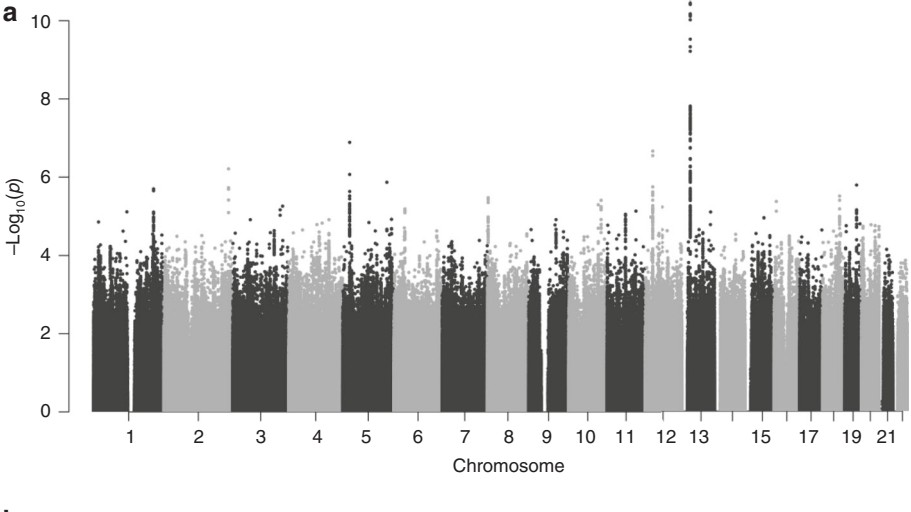

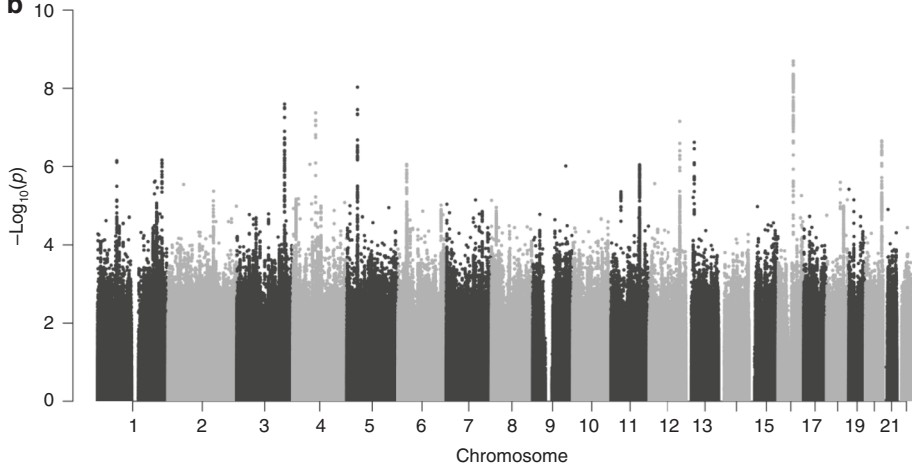

**Fig. 1 Manhattan plots of genome-wide association results from the preeclampsia meta-analyses.** *P*-values ($-\log_{10}$) from the meta-analysis are plotted against their respective positions on each chromosome. **a** Offspring of preeclamptic pregnancies from Europe and Central Asia (6775 cases and 375,372 controls). **b** Preeclamptic women from Europe and Central Asia (9515 cases and 157,719 controls).

$P = 5.3 \times 10^{-7}$ and rs419076 in *MECOM* $P = 7.7 \times 10^{-6}$ (Table 2 and Supplementary Data 3). These four associations are significant after adjusting for the number of variants tested ($P < 0.05/896 = 5.6 \times 10^{-5}$). We thus conclude that in addition to the variants near *ZNF831* and *FTO* that exhibited genome-wide significance, BP variants at *FGF5*, *SH3B2*, and *MECOM* also associate with preeclampsia.

The *ZNF831* variants rs259983 and rs6015450, identified through the preeclampsia meta-analysis and thorough testing of BP variants, respectively, are in strong LD ($r^2 = 0.74$ in European data; Supplementary Table 5). Furthermore, the variants at *FGF5*, *SH3B2,* and *MECOM* that we followed up from our meta-analysis are in LD with the corresponding proximal BP variants (Table 2 and Supplementary Data 2, Supplementary Table 5, Supplementary Figs. 3 and 4). The lowest LD was between the two *MECOM* variants ($r^2 = 0.36$ in European data and 0.12 in Kazakh data) but the *FGF5* and *SH3B2* variant pairs were in strong LD ($r^2 > 0.8$; Supplementary Table 5).

We further compared the effect estimates of the 896 BP variants[16] on preeclampsia in our meta-analysis, with the effect on non-pregnancy diastolic and systolic BP and hypertension based on meta-analyses of Icelandic and UK Biobank (UKBB) data (Supplementary Fig. 5). We found that the four preeclampsia associated variants at *MECOM*, *FGF5*, *ZNF831,* and *SH2B3* are among the variants with the highest effect on all three BP traits, particularly on diastolic BP (Supplementary Fig. 5).

The 892 BP variants that did not individually show significant association with preeclampsia, did, however, show concordance of the preeclampsia risk allele with higher BP allele in meta-analyses of European mothers (588 variants concordant, binomial test $P < 1.2 \times 10^{-21}$), Central Asian mothers (504 variants concordant, binomial test $P < 2.9 \times 10^{-5}$) and in European and Central Asian mothers combined (591 variants concordant, binomial test $P < 1.1 \times 10^{-22}$) (Supplementary Table 6 and see "Methods" section).

**Effect of maternal and fetal genomes**. Through analyses of fetal and maternal genomes, we have identified independent loci that associate with preeclampsia. Since mother and fetus share half their genomes, any variant that associates with preeclampsia only through the fetus is expected to show half the effect in the mother and vice versa. In our previous study, we showed that the frequency of the risk allele of the index variant at the fetal *FLT1* locus in affected mothers was halfway between that in the offspring and controls[6]. We further showed that the effect of the variant was limited to the fetal genome, with no independent effect of the maternal variant.

In the current meta-analyses, we see an effect of associated variants in both maternal and fetal genomes (Supplementary Table 7). We again disentangle the maternal and fetal effects using EMIM[17] (estimation of maternal, imprinting, and

**Table 1 Association statistics for lead maternal preeclampsia variants.**

| SNP | Locus | EA/OA | EAF$_{EUR}$ | EAF$_{CA}$ | Discovery | | Follow-up | | Combined | |
| | | | | | N = 9515/157,719 | | N = 2635/6379 | | N = 12,150/164,098 | |
| | | | | | OR (95% CI) | P | OR (95% CI) | P | OR (95% CI) | P |
|---|---|---|---|---|---|---|---|---|---|---|
| rs259983 | ZNF831 /20q13 | C/A | 0.15 | 0.08 | 1.15 (1.09–1.22) | $2.2 \times 10^{-7}$ | 1.25 (1.11–1.40) | $1.4 \times 10^{-4}$ | 1.17 (1.11–1.23) | $2.9 \times 10^{-10}$ |
| rs1421085 | FTO /16q12 | C/T | 0.41 | 0.28 | 1.13 (1.08–1.17) | $2.0 \times 10^{-9}$ | 1.06 (0.99–1.14) | 0.089 | 1.11 (1.07–1.15) | $1.2 \times 10^{-9}$ |

Locus refers to the nearest gene and chromosomal location.
EA effect allele, OA other allele; EAF$_{EUR}$ and EAF$_{CA}$ effect allele frequency in the meta-analysis of European and Central Asian subjects, respectively, N is the number of individuals in the analysis: cases/controls, OR odds ratio, CI confidence interval, P P-values are two-sided and derived from a fixed-effect meta-analysis of effects and adjusted for genomic control.

**Table 2 Association statistics for blood pressure variants that associate with preeclampsia.**

| SNP[a] | Locus | EA/OA | EAF$_{EUR}$ | EAF$_{CA}$ | Discovery | | Follow-up | | Combined | |
| | | | | | N = 9515/157,719 | | N = 2635/6379 | | N = 12,150/164,098 | |
| | | | | | OR (95% CI) | P | OR (95% CI) | P | OR (95% CI) | P |
|---|---|---|---|---|---|---|---|---|---|---|
| rs1918975 | MECOM /3q26 | T/C | 0.6 | 0.61 | 1.12 (1.07–1.16) | $2.5 \times 10^{-8}$ | 1.06 (0.99–1.14) | 0.094 | 1.10 (1.07–1.14) | $1.2 \times 10^{-8}$ |
| rs419076[b] | MECOM /3q26 | T/C | 0.47 | 0.32 | 1.09 (1.05–1.13) | $7.7 \times 10^{-6}$ | | | | |
| rs1458038 | FGF5 /4q21 | T/C | 0.32 | 0.37 | 1.12 (1.07–1.16) | $4.2 \times 10^{-8}$ | 1.07 (1.00–1.15) | 0.063 | 1.11 (1.07–1.15) | $1.2 \times 10^{-8}$ |
| rs16998073[b] | FGF5 /4q21 | T/A | 0.31 | 0.34 | 1.12 (1.07–1.17) | $8.8 \times 10^{-8}$ | | | | |
| rs10774624 | SH2B3 /12q24 | G/A | 0.43 | 0.22 | 1.11 (1.08–1.16) | $7.0 \times 10^{-8}$ | 1.08 (0.99–1.17) | 0.072 | 1.11 (1.07–1.15) | $1.7 \times 10^{-8}$ |
| rs3184504[b] | SH2B3 /12q24 | T/C | 0.44 | 0.22 | 1.11 (1.07–1.16) | $5.3 \times 10^{-7}$ | | | | |

Locus refers to the nearest gene and chromosomal location.
EA effect allele, OA other allele; EAF$_{EUR}$ and EAF$_{CA}$ effect allele frequency in the meta-analysis of European and Central Asian subjects, respectively, N is the number of individuals in the analysis: cases/controls, OR odds ratio, CI confidence interval, P P-values are two-sided and derived from a fixed-effect meta-analysis of effects and adjusted for genomic control.
[a]Correlation ($r^2$) between variants in European and Kazakh data: chr3: 0.36 and 0.12; chr4: 0.94 and 0.83; chr12: 0.88 and 0.97, respectively.
[b]Lead BP variants at each locus as specified in Evangelou et al.[16].

interaction effects using multinomial modeling; see "Methods" section and Supplementary Note 2). We performed EMIM analysis on the sentinel variants at each of the five maternal loci using genotypes of trios and mother-baby duos that were available in two of the contributing studies. We found no independent effect of the fetal genome for any of the variants after accounting for the maternal genome. With the exception of the *ZNF831* variant rs259983, the association was significant for each maternal variant after accounting for the fetal effect (Supplementary Data 4).

**Preeclampsia stratified by onset.** Early-onset preeclampsia, often clinically defined as onset before 34 weeks gestation, is associated with more severe maternal and fetal complications. We therefore tested the association of our lead maternal and fetal variants with preeclampsia stratified by gestation at diagnosis of disease in four European data sets of 1797 and 3757 early and late-onset maternal cases and 800 and 2660 early and late-onset offspring, respectively (Supplementary Table 8). Consistent with our previous report[6], the effect of the fetal *FLT1* variant rs4769612 was mostly seen in late-onset cases, with a significant difference between early and late-onset cases in a case/case analysis ($P = 0.014$, Table 3). For one of the maternal discovery variants, rs10774624 near *SH2B3*, the effect in early-onset cases was larger than in late-onset cases ($P = 0.016$) (Table 3). For the other four maternal variants, the effect size was not different (Table 3).

**Effect on birth weight.** Preeclampsia is frequently associated with fetal growth restriction and low birth weight. We therefore made use of the large UKBB data set, which includes self-reported birth weight, to assess the effect of the variants on birth weight, regardless of preeclampsia status. The fetal preeclampsia *FLT1* variant was tested using the individual's own birth weight ($N = 236,507$) while the maternal preeclampsia variants were tested using the reported birth weight of the female participant's first child ($N = 178,241$). The fetal *FLT1* variant was associated with lower birth weight ($P = 2.6 \times 10^{-3}$, Table 3) and two of the maternal risk variants rs1918975 at *MECOM* and rs10774624 at *SH2B3* associated with lower birth weight of first child with $P = 1.9 \times 10^{-4}$ and $P = 4.5 \times 10^{-22}$, respectively (Table 3). The BP locus at *SH2B3* has been shown to associate with maternal effect on birth weight[18], and consistent with our data the BP raising allele associated with lower birth weight. In contrast, we find no evidence in this population data set for the association of the *FGF5*, *ZNF831*, and *FTO* loci with the birth weight of first child.

**Heritability of preeclampsia.** We made use of the genetic data to study the heritability of preeclampsia. We applied GCTA Genomic Relatedness Restricted Maximum Likelihood (GREML) analysis to the chip genotypes of European and Central Asian subjects to estimate the SNP heritability of preeclampsia on the liability scale[19] (see "Methods" section). For Europe, we found the heritability for maternal preeclampsia to be 38.1% (95% CI: 29.3–46.8) and in fetal preeclampsia 21.3% (95% CI: 7.4–35.3); for Central Asia, the heritability was found to be 54.4% (95% CI: 29.6–79.3) and 42.5% (95% CI 17.3–67.7) for maternal and fetal preeclampsia, respectively (Supplementary Table 9). These results are consistent with those previously reported in European family-based studies[5].

**Gestational hypertension.** Gestational hypertension is defined as new-onset hypertension at or after 20 weeks gestation in the absence of proteinuria. This condition is not commonly associated with maternal and perinatal mortality, although it is linked

**Table 3 Effect of preeclampsia variants on disease onset and birth weight.**

| Variant | Locus | RA/OA | Early-onset N_off = 800/378,185; N_mat = 1797/143,233 | | Late-onset N_off = 2660/377,382; N_mat = 3757/144,355 | | Early vs late-onset N_off = 800/2660; N_mat = 1797/3757 | | Birth weight N_off = 236,507; N_mat = 178,241 | |
|---|---|---|---|---|---|---|---|---|---|---|
| | | | OR (95% CI) | P | OR (95% CI) | P | OR (95% CI) | P | Effect (g) (95% CI) | P |
| Offspring | | | | | | | | | | |
| rs4769612 | FLT1 /13q12 | C/T | 1.01 (0.90–1.14) | 0.83 | 1.20 (1.12–1.30) | $8.1 \times 10^{-7}$ | 0.84 (0.74–0.97) | 0.014 | −6.0 (−9.9, −2.0) | $2.6 \times 10^{-3}$ |
| Maternal | | | | | | | | | | |
| rs259983 | ZNF831 /20q13 | C/A | 1.12 (1.01–1.25) | 0.039 | 1.14 (1.06–1.23) | $6.8 \times 10^{-4}$ | 0.99 (0.90–1.09) | 0.84 | 1.6 (−3.8, 7.1) | 0.57 |
| rs1421085 | FTO /16q12 | C/T | 1.15 (1.06–1.24) | $4.0 \times 10^{-4}$ | 1.11 (1.05–1.17) | $3.2 \times 10^{-4}$ | 1.01 (0.94–1.08) | 0.77 | 2.7 (−1.6, 7.1) | 0.21 |
| rs1918975 | MECOM /3q26 | T/C | 1.16 (1.07–1.25) | $2.1 \times 10^{-4}$ | 1.11 (1.05–1.18) | $6.0 \times 10^{-4}$ | 1.03 (0.94–1.13) | 0.53 | −7.6 (−11.4, −3.8) | $1.9 \times 10^{-4}$ |
| rs1458038 | FGF5 /4q21 | T/C | 1.14 (1.05–1.24) | $1.8 \times 10^{-3}$ | 1.12 (1.05–1.19) | $2.6 \times 10^{-4}$ | 1.02 (0.92–1.13) | 0.7 | −1.6 (−5.4, −2.2) | 0.41 |
| rs10774624 | SH2B3 /12q24 | G/A | 1.23 (1.13–1.33) | $6.2 \times 10^{-7}$ | 1.09 (1.03–1.16) | $4.5 \times 10^{-3}$ | 1.12 (1.02–1.23) | 0.016 | −19.6 (−23.4, −15.8) | $4.5 \times 10^{-22}$ |

Association of the fetal variant rs4769612 was tested in preeclampsia offspring case-control data, early versus late-onset case-control data, and in birth weight data from a set of 236,507 subjects from the UKBB; association of maternal variants was tested in maternal preeclampsia early and late-onset case-control data, early versus late-onset case-control data, and in data on birth weight of the first child as reported by 178,241 mothers from the UKBB. Birth weight effect is reported in grams. Locus refers to the nearest gene and chromosomal location. Significance threshold: $P = 0.05/6 = 0.0083$. P-values for effects on disease onset are obtained from fixed-effect meta-analysis of effects, adjusted for genomic control. P-values for effects on birth weight are obtained from linear regression of birth weight on genotype count, adjusting for covariates (see "Methods" section). All P-values are two-sided.
RA risk allele, OA other allele, N_off is the number of individuals in the offspring analysis: cases/controls, N_mat is the number of individuals in the maternal analysis: cases/controls. OR odds ratio, CI confidence interval.

to lower birth weight and subsequent maternal cardiovascular disease. We explored the relationship between gestational hypertension and the individual variants that we showed are associated with preeclampsia, by testing their association in small data sets of 4275 offspring and 3428 maternal cases from pregnancies diagnosed with gestational hypertension (Supplementary Table 1). The risk allele of the FLT1 variant did not associate with increased risk in gestational hypertension offspring ($P = 0.21$) (Supplementary Table 10). The maternal variants at FTO and SH2B3 both associated with gestational hypertension in maternal cases with $P = 1.7 \times 10^{-3}$ and $3.7 \times 10^{-5}$, respectively, while the other preeclampsia variants did not associate with gestational hypertension (Supplementary Table 10). However, all five maternal preeclampsia variants had the same risk allele as the gestational hypertension maternal variant. As these risk alleles also increase BP, we looked at the other 892 BP variants, previously analyzed for preeclampsia, and found excess of concordance between BP-increasing alleles and gestational hypertension risk alleles (564 variants concordant, binomial test $P < 5 \times 10^{-16}$) (Supplementary Table 6 and see "Methods" section).

**Genetic overlap with other traits**. In addition to hypertension, epidemiological evidence indicates a shared risk between preeclampsia and several vascular, metabolic, and inflammatory traits[20–23]. Using the cross-trait LD-score regression method[24,25], we estimated the genetic correlation between preeclampsia, based on the maternal meta-analysis, and a selection of 12 relevant traits in deCODE and UKBB data (see Methods). We found a positive genetic correlation between preeclampsia and both systolic and diastolic BP, hypertension, CAD and T2D and a negative correlation with birth weight of first child (Fig. 2 and Supplementary Table 11).

Genetic correlation between preeclampsia and BP is expected and in line with our finding of BP variants associating with preeclampsia. Correlation between preeclampsia and the risk of developing T2D has been noted previously and is consistent with the association between preeclampsia and gestational diabetes[26]. In our data, the genetic correlation of preeclampsia with T2D is second only to its correlation with hypertension and BP.

We used a polygenic risk score (PRS) analysis to explore further the correlation between preeclampsia and hypertension, T2D, CAD, BMI, and birth weight. To avoid confounding by population structure, we extracted risk alleles, P-values and effect estimates for each trait from studies that did not include Icelandic subjects and used these to calculate a standardized PRS for the Icelandic preeclampsia data set. Consistent with the results from the genetic correlation analysis the PRS for hypertension showed the most significant association with preeclampsia ($P = 1.2 \times 10^{-12}$, effect (log$_e$ odds ratio (OR)) $= 0.18$, 95% CI: 0.13–0.23) (Supplementary Table 12); the effect corresponds to the increase in the risk of preeclampsia per standard deviation in PRS. The PRSs for T2D, BMI and CAD also associated with preeclampsia (Supplementary Table 12). Notably, when we adjusted the T2D, BMI and CAD PRSs for the HT-PRS they were no longer significant, indicating that this association was at least partly due to the increased risk of hypertension implicit in those risk scores (Supplementary Table 12). The PRS for higher birth weight of first child associated with lower risk of preeclampsia and conditioning on the HT-PRS had little effect on this association (Supplementary Table 12) indicating that factors other than hypertension account for the association between birth weight and preeclampsia. We further tested the same risk scores on the Icelandic gestational hypertension data set (Supplementary Table 1). As for preeclampsia, the strongest result was obtained for the HT-PRS ($P = 2.1 \times 10^{-35}$),

but the effect on gestational hypertension was nearly double that on preeclampsia (log$_e$ OR = 0.32, 95% CI: 0.27–0.38) (Supplementary Table 12).

**Hypertension and preeclampsia in Europe and Central Asia**. Deciphering the relationship between preeclampsia and hypertensive disease in pregnancy is of importance to increase our understanding of the causes of preeclampsia. To further explore this relationship, we compared the effects of HT-PRS on three hypertension-related traits, preeclampsia, gestational hypertension, and essential hypertension in females, within the Icelandic population. In these analyses, we restricted the control group to females who had not been diagnosed with hypertensive disorders. The risk score was associated with all three traits (Fig. 3). We note that the removal of hypertensive women from the control group increased the effect of the association between the HT-PRS and preeclampsia (log-odds = 0.18, 95% CI: 0.13–0.23, see Supplementary Table 12, to log-odds = 0.23, 95% CI: 0.19–0.28, Fig. 3). Of note, the effect of the PRS on gestational hypertension was similar to the effect on hypertension and around 1.7-fold higher than the effect on preeclampsia (Fig. 3). Furthermore, the HT-PRS explained 2.4% of the variance in hypertension and only 0.67% of the variance in preeclampsia (Fig. 3). The corresponding genetic risk score based on the five variants discovered here to associate with preeclampsia through the maternal genome only explains 0.25% of the variance in preeclampsia ($P = 1.6 \times 10^{-8}$). The effects of HT-PRS on preeclampsia in two additional European preeclampsia data sets (FINRISK and MoBa) and in the two Central Asian data sets were comparable to the deCODE set (Fig. 3).

## Discussion

We present here a large-scale meta-analysis of maternal preeclampsia as well as an expansion of our previous study of the contribution of the fetal genome to preeclampsia. In addition to European data sets, we also include data from Central Asian populations, previously under-represented in human genetic studies. To facilitate the use of Central Asian subjects in association analysis, we generated a haplotype reference panel based on whole-genome sequencing of individuals from Kazakhstan and Uzbekistan.

Even though we more than doubled the number of cases from our previous meta-analysis of preeclampsia offspring[6] the only significant locus remained the previously reported FLT1 locus. These additional data, however, allowed us to improve the fine mapping of the locus, revealing a third independent signal. The lack of evidence for more loci indicates that any additional variants that associate with preeclampsia through the fetal genome are likely to have less effect and/or lower minor allele frequency and thus they require larger data sets for detection of association (Supplementary Fig. 2).

This study identifies five maternal sequence variants that associate with development of preeclampsia. While only two loci were genome-wide significant in our meta-analysis, three additional loci were identified as significantly associated with preeclampsia through analysis of variants associated with BP. Four of the associated variants are among the strongest BP signals found in a recent meta-analysis of BP traits[16] and the fifth, the FTO variant, is the variant that most strongly associates with BMI in addition to its effect on BP[13,14]. The results of the PRS analysis suggest that the effect of genetic predisposition to BMI on the risk of preeclampsia is mostly through the effect of BMI on BP. This is in contrast to the effect of the FTO variant on preeclampsia, which is disproportionate to its effect on BP.

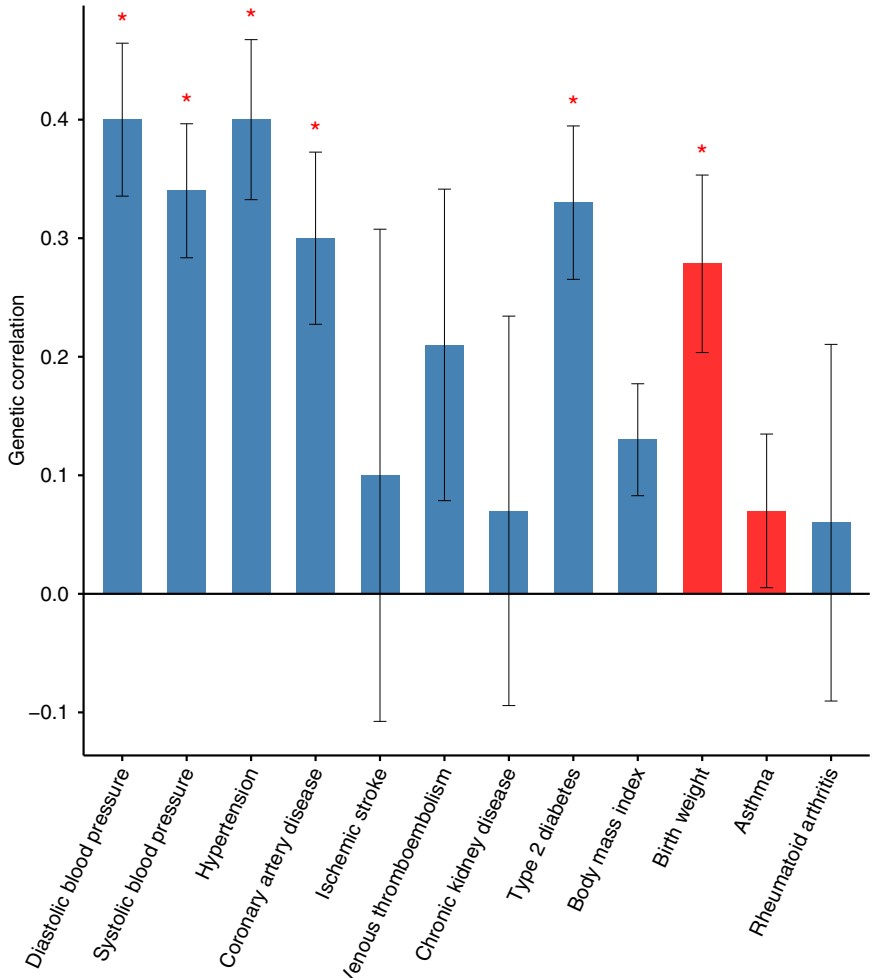

**Fig. 2 Genetic correlation between maternal preeclampsia and selected traits.** Genetic correlation between pairs of traits using the cross-trait LD-score regression method in the European maternal preeclampsia data sets and the summary statistics from deCODE and UK Biobank data sets for each secondary trait. On one hand, we calculated the genetic correlation between preeclampsia meta-analysis of GOPEC, ALSPAC, and MoBa data, and deCODE GWAS summary statistic for each secondary trait, and on the other hand between preeclampsia meta-analysis of deCODE, SSI, and FINRISK data, and UK Biobank GWAS summary statistic for each secondary trait. The P-values and genetic correlation estimates presented are the meta-analysis of the two independent tests, with the exception of birth weight of first child where the results only include the preeclampsia meta-analysis excluding UK data sets and the UK biobank data on birth weight of the first child. The height of the bars indicates the genetic correlation, blue bars indicate a positive correlation, red bars indicate a negative correlation. Error bars indicate standard error. Red asterisks indicate results that are significant after accounting for the 12 traits tested. Significance threshold: P = 0.05/12 = 0.0042. Details of the results presented in the figure are reported in Supplementary Table 11.

A significant association between HT-PRS and preeclampsia shows that genetic predisposition to hypertension is a major risk factor for preeclampsia. However, it is interesting that whilst the effect of the association with gestational hypertension coincides with the association with hypertension, the effect on preeclampsia is somewhat lower. While these results show an overlap in genetic susceptibility to these two hypertensive disorders of pregnancy they also highlight important differences. In particular, they imply that, although genetic predisposition to hypertension contributes significantly to the risk of preeclampsia, additional factors are involved.

Although the effect estimates of the hypertension PRS were derived from a European data set, association with preeclampsia is similar in the European and Central Asian data indicating that the risk score is measuring the same underlying risk in both populations.

Women with hypertensive pregnancy disorders including preeclampsia have significantly increased risk of future cardiovascular disease[27–29]. Potential explanations include common risk

factors in addition to the vascular damage sustained by the hypertensive episode during pregnancy. Our data show that preeclamptic women have an underlying predisposition to higher BP, which helps explain the risk of cardiovascular disease. Any role of endothelial dysfunction due to circulating inflammatory and anti-angiogenic factors in pregnancy in the causality of future cardiovascular disease remains undetermined. Importantly, our data affirm the diagnosis of hypertensive disease in pregnancy as predictors of future cardiovascular risk in women.

Given the results of the current study, it is likely that a large number of sequence variants together affect the genetic predisposition to preeclampsia, each contributing a small effect through regulation of BP and probably, through other pathways. Such factors could act either through the maternal or fetal genome or both. Revealing those variants will require much larger studies than this current meta-analysis. Importantly, in order to disentangle the maternal and fetal contribution, both mother and child need to be included in future studies of preeclampsia, as highlighted by the *FLT1* locus that is among the most significant

| Study | Cases | Controls | Effect | 95% CI | P–value | $R^2$ |
|---|---|---|---|---|---|---|
| **deCODE** | | | | | | |
| Hypertension | 21,677 | 50,943 | 0.37 | (0.35–0.39) | $1 \times 10^{-300}$ | 0.024 |
| Gestational hypertension | 1532 | 50,943 | 0.39 | (0.34–0.45) | $9.4 \times 10^{-50}$ | 0.018 |
| Preeclampsia | 1662 | 50,943 | 0.23 | (0.19–0.28) | $1.2 \times 10^{-20}$ | 0.0067 |
| **MoBa** | | | | | | |
| Preeclampsia | 1386 | 930 | 0.27 | (0.19–0.36) | $1.6 \times 10^{-10}$ | 0.024 |
| **FINRISK** | | | | | | |
| Preeclampsia | 400 | 7,805 | 0.19 | (0.096–0.29) | $2.6 \times 10^{-4}$ | 0.0050 |
| **Kazakhstan** | | | | | | |
| Preeclampsia | 923 | 864 | 0.27 | (0.17–0.36) | $8.8 \times 10^{-8}$ | 0.022 |
| **Uzbekistan** | | | | | | |
| Preeclampsia | 810 | 884 | 0.17 | (0.08–0.27) | $4.9 \times 10^{-4}$ | 0.009 |
| **META preeclampsia** | **5,181** | **61,426** | **0.23** | **(0.20–0.27)** | $1.1 \times 10^{-40}$ | |

Effect (95% CI) 0.1 0.2 0.3 0.4

**Fig. 3 Polygenic risk score analysis using PRS for hypertension.** PRS effect estimates were based on GWAS analysis of the UKBB hypertension data set. The top panel shows association between the HT-PRS and hypertension (females only), gestational hypertension and preeclampsia in genotyped subjects from the deCODE cohort. The control group comprises females that are not on any of the case lists (hypertension-free controls). For other studies, the association analysis used the preeclampsia cases and controls that were included in the respective maternal GWAS analyses. Effect reported as log-odds corresponds to the increase in risk of the respective trait for one standard deviation of the hypertension risk score. 95% CI: 95% confidence interval. P-values are obtained from logistic regression of case status on individuals' polygenic risk score, adjusted for covariates (see "Methods" section). All P-values are two-sided. $R^2$ denotes the explained variance.

signals in our maternal association study even though its effect on susceptibility arises only from the fetal genome.

## Methods

**Participants, genotyping, imputation, and association analysis**. The combined fetal meta-analysis included eight data sets with a total of 6775 cases and 375,372 controls. The combined maternal meta-analysis included 9515 cases and 157,719 controls. Both studies were based on subjects of Northern European and Central Asian origin. A further 2300 maternal cases and 5325 controls were used to follow-up variants with a suggestive association ($P < 1 \times 10^{-6}$) in the meta-analysis. An overview of the data sets is shown in Supplementary Table 1 and an overview of clinical details for each data set is reported in Supplementary Table 13. Written informed consent for genetic studies was obtained from participants, or from parents on behalf of minors, and all studies were approved by local Research Ethics Committees.

The GOPEC cohort comprises 1875 white western European women with preeclampsia and 1004 offspring recruited in the UK for a number of genetic studies of preeclampsia conducted between 1989 and 2010. 72% of women were recruited at diagnosis[30,31], 11% were recruited prospectively from pregnancy cohorts[32], and 17% were identified from the Aberdeen Maternity and Neonatal Databank[33]. Women affected by preeclampsia were selected for this study based on an internationally recognized definition[34]: new-onset hypertension after the 20th week of gestation, with systolic BP ≥ 140 mmHg or diastolic BP ≥ 90 mmHg on at least two occasions; and new-onset proteinuria of 0.3 g/24 h or more, or ≥1+ on dipstick analysis of urine. All were singleton pregnancies. Exclusion criteria included pregnancies in women with a previous history of essential hypertension, type 1 or type 2 diabetes or chronic renal disease.

All subjects provided a sample of venous blood for DNA extraction. Offspring from preeclamptic pregnancies were available from two cohorts recruited at diagnosis; DNA was extracted from the umbilical cord or capillary blood spots obtained for neonatal screening.

Control data were derived from the WTCCC2 genome-wide analysis of 2930 samples from the 1958 Birth Cohort and 2737 samples from the National Blood Services, providing control data for 5297 individuals after quality control (QC)[35]. Details of genotyping and imputation of the cases and controls have been reported[6,36] and the number of subjects included in each analysis is shown in Supplementary Table 1. The study was approved by the Derbyshire Research Ethics Committee.

The deCODE preeclampsia cohort is a part of an ongoing sample collection including a large fraction of the Icelandic population[6]. Briefly, singleton affected pregnancies were identified through the scrutiny of hospital records from 1970 to 2017 at Landspitali University Hospital in Reykjavik. Records from 1999 and earlier were examined and each affected pregnancy reclassified while pregnancies from 2000 to 2017 were identified based on ICD-10 codes O14; O15 for preeclampsia and O13 for gestational hypertension. Further data on each pregnancy, including gestational duration at diagnosis of preeclampsia and BMI, were obtained directly from maternity records. Genotyping and imputation methods in the Icelandic samples were essentially as described previously[37,38]. In short, we sequenced the whole genomes of 15,220 Icelanders using Illumina technology to a mean depth of at least 10× (median 32×). SNPs and insertions and

deletions (indels) were identified and their genotypes called using joint calling with the Genome Analysis Toolkit HaplotypeCaller (GATK version 3.4.07)[39]. Information about haplotype sharing was used to improve genotype calls, taking advantage of the fact that all of the sequenced individuals had also been chip-typed and long-range phased. The 33.4 million variants that passed the high-quality threshold were then imputed into 151,677 Icelanders who had been genotyped with various Illumina SNP chips and their genotypes phased using long-range phasing[40]. We further increased the sample size for association analysis and thus the power to detect associations by imputing the sequence variants into 282,894 un-typed relatives of the chip-typed individuals using genealogic information. This data set includes the maternal preeclampsia cases and their offspring as well as the age-, sex-, and county-of-origin-matched controls. The control group comprised individuals recruited through different genetic research projects at deCODE. Out of 2389 maternal cases included in the analysis, 1662 were chip-typed and 435 of the 2221 offspring included in the analysis were chip-typed. All of the variants that were tested had imputation information above 0.8. To account for inflation in test statistics due to cryptic relatedness and stratification, we applied the method of LD-score regression[24]. With a set of 1.1 million variants, we regressed the $\chi^2$ statistics from our GWAS scan against the LD-score and used the intercept as a correction factor. LD scores were downloaded from an LD-score database (ftp://atguftp.mgh.harvard.edu/brendan/1k_eur_r2_hm3snps_se_weights.RDS; accessed 23 June 2015). The estimated inflation factors based on LD-score regression were 1.08 for maternal preeclampsia and 1.14 for the offspring analysis. The study was approved by the Icelandic National Bioethics Committee (VSN-14-174). Written informed consent was obtained from all genotyped subjects.

ALSPAC (Avon Longitudinal Study of Parents and Children) is a prospective birth cohort study, which recruited 15,454 pregnant women with expected delivery dates between 1st April 1991 and 31st December 1992 living in and around the city of Bristol in the South West of England. There was a total of 15,589 fetuses. Of these 14,901 were alive at 1 year of age. The study has been described in full elsewhere[41,42]. Obstetric data, including all measurements of BP (median number 12; IQR: 11–16) and of proteinuria (12; 9–14) were extracted from obstetric records by one of six trained midwives. Gestational hypertension was defined as systolic BP ≥ 140 mmHg OR diastolic BP ≥ 90 mmHg on at least two occasions after 20 weeks of gestation in women who had not previously been diagnosed with hypertension outside of pregnancy. Preeclampsia was defined as hypertension as described for gestational hypertension accompanied by at least 1+ proteinuria on dipstick testing (Albustix; Ames Co, Elkhart, Ind.) occurring at the same time as the episodes of raised BP. The comparison (control) group were all other included women or offspring with GWAS data who were not affected by hypertension during pregnancy.

A total of 9912 ALSPAC children were genotyped using the Illumina HumanHap550 quad genome-wide SNP genotyping platform[6]. ALSPAC mothers were genotyped using the Illumina human660W-quad array at Centre National de Génotypage (CNG) and genotypes were called with Illumina GenomeStudio. Details of genotyping and quality control can be found elsewhere[43]. Haplotype phasing was performed using ShapeIT (v2.r644) and known autosomal variants were imputed with Impute V2.2.2 using the 1000 genomes reference panel (Phase 1, Version 3) consisting of 2186 reference haplotypes (including non-Europeans). All imputed dosages converted to best guess genotypes in binary plink format,

using a hard call threshold of 0.1. Cases and controls for gestational hypertension analysis were imputed with phased haplotypes from the Haplotype Reference Consortium (HRC) panel. Ethical approval was obtained from the ALSPAC Ethics and Law Committee and the Local Research Ethics Committees. Please note that the study website contains details of all the data that is available through a fully searchable data dictionary and variable search tool (http://www.bristol.ac.uk/alspac/researchers/our-data/).

The Norwegian MoBa (Mother, Father and Child Cohort Study) is a longitudinal study of over 110,000 pregnant women, their children and partners, recruited between 1999 and 2008 from maternity units throughout Norway, and has been described previously[44]. DNA was extracted manually from whole blood samples obtained at recruitment (around 18 gestational weeks) using the FlexiGene kit (Qiagen, Hilden, Germany). Birth outcome information for all MoBa participants is obtained through linkage with the Medical Birth Registry of Norway (MBRN)[45]. In the MoBa Preeclampsia case–control study, all preeclampsia cases identified through linkage with the MBRN, and a subset of pregnancies unaffected by preeclampsia, were validated using antenatal records and hospital discharge codes[46]. Preeclampsia was defined using American College of Obstetrics and Gynecologists criteria[47]. We included, from among these validated records, women with a singleton pregnancy who conceived spontaneously, were verified cases or controls, returned both early and late pregnancy study questionnaires, had placenta stored in the MoBa biobank, and had no history of chronic hypertension: 1564 validated preeclampsia cases (1118 had both mother and child DNA, and 446 had only maternal DNA) and 999 controls (of which 968 had a mother and child DNA, and 31 had only maternal DNA). The date of onset was defined as the first gestational week when both blood pressure and proteinuria criteria were simultaneously noted in the antenatal chart. Mothers were genotyped by the UNC Mammalian Genotyping Core using the HumanCoreExome-12 Bead Chip from Illumina (Illumina, Inc., San Diego, CA). Samples and SNPs were examined using PLINK 1.07 for quality control. SNPs were excluded if the missing rate exceeded 5%, there was a substantial deviation from HWE ($P < 1 \times 10^{-3}$) or the MAF was <0.125. For each pair of related mothers, we preferentially included the one with the most complete genetic data, or in the case of equivalence, randomly sampled between them. Quantile-quantile plots and calculation of genomic control lambda ($\lambda$GC = 1.01) indicated no systematic test statistic inflation, unidentified relationships, or cryptic admixture. Outliers for any of the first three 1000 Genomes axes of variation (based on CEU, YRI, CHB, PUR, CLM, and MXL) > 3 standard deviations from the mean were excluded. The post-QC data set was imputed using PBWT and pre-phased using SHAPEIT2 against the 1000 Genomes Phase 3 reference panel. Imputation for GWAS analysis was conducted by the Sanger Imputation Service provided by the Wellcome Trust Sanger Institute. Data used for the analysis of disease onset and PRS was imputed at deCODE genetics using the same reference panel. The study was approved by the Regional Committee for Medical Research, South East Norway.

SSI (Statens Serum Institut) Danish study subjects were drawn from a case–control study of severe, early-onset preeclampsia and were all of Scandinavian (Danish, Faroese, Norwegian, Swedish, or Icelandic) ancestry. Severe preeclampsia cases were identified through the Danish National Patient Register, which includes all hospital diagnoses assigned since 1977[48], using the ICD-8 code 637.04 and the ICD-10 codes O14.1, O14.2, and O15.0–15.9. Timing of onset was defined based on gestational age at delivery. Women were classified as having had early preterm preeclampsia if they delivered before 34 completed weeks of pregnancy and late preterm preeclampsia if they delivered between 34 and 36 completed weeks of pregnancy. Controls for the maternal analysis were randomly selected from among parous women without cardiovascular disease, diabetes or kidney disease and with a history of only healthy pregnancies ending in live births at 40 weeks' gestation, and were matched 1:1 to the case women on maternal birth year and age at first delivery. Fetal cases were singletons from preeclamptic pregnancies of a subset of cases from the maternal analysis and controls were individuals genotyped with the same array (Illumina Multi-Ethnic Global v2 A2 at SSI). Biological samples were drawn from the Danish Neonatal Screening Biobank[49] and the biobank of the Danish National Birth Cohort[50], both of which are part of the Danish National Biobank. This study was assessed by the Scientific Ethics Committee of the Danish Capital City Region (Copenhagen) (approval no. H-6-2013-008) and the Danish Data Protection Agency (approval no. 2015-57-0102). The Danish Scientific Ethics Committee granted an exemption from obtaining informed consent from study participants as this research project was based on existing biological samples already held in a biobank. All SSI samples were genotyped using the Illumina Multi-Ethnic Global v2 A2 array. Data cleaning and quality control were performed using a sequential procedure. After the quality control steps, we ended up with 872 cases and 815 controls for the maternal analysis and 213 cases and 963 controls for the fetal analysis. After genotyping QC, we imputed unobserved genotypes with phased haplotypes from the HRC panel (version r1.0). Association analysis of imputed autosomal SNPs with preeclampsia was performed using logistic regression on imputed SNP dosages that had an imputation info score ≥ 0.8 and MAF > 1%, under an additive genetic model using PLINK[51].

FINRISK is a series of health examination surveys carried out by the National Institute for Health and Welfare of Finland every 5 years in 1972–2012 and has been described previously[52]. The surveys are based on random population samples from five (six in 2002) geographical regions of Finland. The age-range was 25 to 64 years until 1992 and 25 to 74 years since 1997. The survey includes a self-administered questionnaire, a standardized clinical examination carried out by specifically trained study nurses and the collection of a blood sample for laboratory measurements and DNA extraction[53]. We chose 402 women with a history of preeclampsia and 7924 female controls with normal delivery from the 1992, 1997, 2002, 2007, and 2012 FINRISK surveys. For identifying preeclamptic and eclamptic women, we used the following Finnish International Classification of Disease (ICD) codes in the comprehensive National Hospital Discharge Register covering years 1992 to 2007: ICD-10 (in use since 1996): O14.0, O14.1, O14.9, O15.0, O15.1, O15.2, O15.9; ICD-9 (in use from 1987 to 1996): 6424 to 6426, 6427A; and ICD-8 (in use from 1968 to 1986): 637.03, 637.04, 637.09, 637.10, 637.99. The FINRISK samples were genotyped using multiple different genotyping chips, for which the QC, phasing, and imputation were done in multiple chip-wise batches. Principal component (PC) analysis was applied to detect population structures and outliers removed. Imputation was done utilizing a population-specific reference panel (http://www.sisuproject.fi) of 2690 high-coverage whole-genome and 5,093 high-coverage whole-exome sequences with IMPUTE2[54]. The FINRISK study was approved by the Helsinki University Hospital ethical committee.

Central Asian subjects of Uzbek and Kazakh ancestry, determined by grandparental ethnicity, were recruited between 2012 and 2015 from six maternity units in Uzbekistan and seven in Kazakhstan. Women with singleton pregnancies affected by preeclampsia were recruited at the time of diagnosis; healthy pregnant controls were recruited from the same maternity centers. The diagnostic criteria for preeclampsia were: systolic BP ≥ 140 mmHg and diastolic BP ≥ 90 mmHg on at least two occasions within 24 h after the 20th week of pregnancy in a previously normotensive woman, accompanied by proteinuria ≥300 mg/L. Women below the age of 18, those with a prior history of hypertension, non-infective renal disease, type 1 or type 2 diabetes, or 3 or more consecutive miscarriages, were excluded. Volunteers provided a sample of venous blood for DNA extraction. DNA from the offspring of preeclamptic pregnancies was extracted from a sample of umbilical venous blood or umbilical cord obtained at delivery. All volunteers gave informed consent for participation in the study; mothers gave consent on behalf of their offspring.

Three batches of Kazakh preeclamptic mothers, babies, and control sets were genotyped at the deCODE genotyping facility using Illumina SNP genotyping platforms. Kazakh 1, including 1003 sets was genotyped on Illumina 2.5-8, Kazakh 2, including 761 set, on OmniExpress, and finally Kazakh 3, including 772 sets, on Infinium GSA. We carried out quality control analysis using PLINK (http://zzz.bwh.harvard.edu/plink/), SMARTPCA (https://www.hsph.harvard.edu/alkes-price/software/Eigensoft) and Admixture (http://dalexander.github.io/admixture/).

For each batch variants with call rate <95% and Hardy–Weinberg equilibrium $P < 1 \times 10^{-6}$ were excluded. QC tests on the genotyped samples included yield, parent–child relationship and gender test. Samples with call rate <95% were excluded from the study. Duplicate samples were excluded except where the parent–child relationship confirmed the correct identity of the sample. Male samples from maternal and control groups as well as all samples in the control group with a genotyped first-degree relative were excluded. In order to account for population structure, we applied PC analysis and excluded population outliers from the study. PC analysis was performed separately on each association data set. The final number of samples included in each analysis is shown in Supplementary Table 1. Over 70% of the original samples were included. Exclusion of population outliers removed 10% of the samples, mostly due to exclusion of 28% of all control samples.

For each of the three batches, maternal and fetal case–control sets were separately imputed using IMPUTE2. The reference panel consisted of a Central Asian whole-genome sequence panel (see Supplementary Note 1) merged with 1000 genomes Phase 3 haplotype reference panel. The Ministry of Health, Republic of Kazakhstan, Central Ethics Committee gave approval for the conduct of the study.

2869 samples from Uzbek preeclamptic mothers, their offspring, and controls were genotyped on the Omni 2.5-8 chip. The genotypes were called using Gencall genotype calling algorithm. Quality control measures included gender and relatedness checks, and samples with call rates <95% or heterozygosity >± 3 SD were excluded, resulting in the retention of 2742 samples. The quality of the calls for individual variants was assessed by looking at founder individuals (mothers and controls). The following QC filters were applied to autosomal SNPs using the plink software (version 1.90b downloaded from www.cog-genomics.org/plink2): violation of Hardy–Weinberg Equilibrium; haplotype-based test for non-random missing data, Bonferroni corrected $P < 0.05$; call rate <98%. A total of 2,292,786 autosomal variants were genotyped. The quality control procedure identified 50,793 poorly genotyped SNPs leaving 2,241,993 SNPs available for further analysis.

The GWAS samples were pre-phased using SHAPEIT and then imputed using IMPUTE2. The Uzbek samples were imputed using the reference panel described for imputation of Kazakh samples, constructed from a Central Asian whole-genome sequencing panel merged with 1000 Genomes Phase 3. We filtered the imputed variants by requiring that the info >0.8. In total, >11 million imputed and genotyped variants were available for further analysis. The study was approved by the National Ethics Committee, Ministry of Health, Republic of Uzbekistan.

**Association analysis.** Logistic regression was used to test for association between variants and disease, assuming an additive model, treating disease status as the

response and expected genotype counts from imputation as covariates. For the deCODE cohort, information on the county of origin within Iceland were included as covariates to adjust for possible population stratification. This was done using software developed at deCODE genetics[37]. For the SSI cohort association, analysis was done using PLINK. GOPEC, ALSPAC, MoBa and Uzbek cohorts included the top five ancestry principal components as covariates (SNPTEST (v2.4.1)). For the FINRISK and Kazakh cohorts, the top twenty (FINRISK) and ten (Kazakhstan) ancestry principal components were included as covariates (SNPTEST (v2.5)[55].

**Follow-up data sets.** *FINNPEC (Finnish Genetics of Preeclampsia Consortium):* The FINNPEC collection was assembled in Finland between 2008 and 2011 from two recruitment arms and has been described previously[56]. Samples were collected at the time of diagnosis of preeclampsia from 879 mothers and during pregnancy from 922 non-preeclamptic mothers from antenatal and labor wards. Their children and partners were also enrolled. A further 525 pregnancies affected by pre-eclampsia were identified by examination of hospital records, and women and offspring were invited to participate by letter. After exclusion of pregnancies that did not meet the entry criteria for this study, 678 preeclamptic mothers and 580 offspring were included as cases, and 700 non-hypertensive mothers and 760 offspring provided the control group. Offspring genotype data included in this study were generated in our previous study[6]. Samples were genotyped at the BHF Glasgow Centre for Cardiovascular Research using an OpenArray platform, and at the Wellcome Trust Sanger Institute using Sequenom technology. The Hospital District of Helsinki and Uusimaa Co-ordinating Ethics Committee approved the study.

*DNBC (Danish National Birth Cohort):* The Danish follow-up samples were drawn from the DNBC, a population-based cohort of more than 100,000 pregnancies, recruited in the years 1996–2002, and have been described previously[50]. Extensive phenotype information is available for the DNBC mothers and children based on computer-assisted telephone interviews, questionnaire-based follow-up surveys and data from the Danish population and health registers. DNBC women with preeclampsia were identified from the Danish National Patient Register[48] using ICD-8 codes 63700-63719 and ICD-10 code groups O13.9, O14 and O15. Only singleton pregnancies were included. As controls, we used mothers without any preeclampsia-related diagnosis codes in any pregnancy. The Scientific Ethics Committee for the Capital City Region (Copenhagen) and the Danish Data Protection Agency approved the study. The Scientific Ethics Committee also granted exemption from obtaining informed consent from participants (H-B-2007-124) as the study was based on biobank material. Follow-up variants were genotyped using the Centaurus (Nanogen)[57] or KASP (LGC Genomics) platform.

*The HUNT study* (HUNT) is an ongoing longitudinal health survey of the population of Nord-Trøndelag County in Norway, including ~120,000 individuals[58]. DNA samples are available from ~70,000 subjects recruited to HUNT2 and HUNT3, in 1995–1997 and 2006–2008, respectively. Women with a history of preeclampsia were identified retrospectively by linking the HUNT database to the Medical Birth Registry of Norway (MBRN), using diagnosis codes ICD-8 (before 1998) and ICD-10 (after 1998). The MBRN defines preeclampsia as an increase in BP to at least 140 systolic or 90 mmHg diastolic (or an increase in diastolic BP ≥ 15 mmHg from the level measured before 20th gestational week), combined with proteinuria (protein excretion of ≥0.3 g per 24 h or ≥1+ on dipstick). These criteria are in accordance with the diagnostic criteria used clinically in Norway until 2006. Of the HUNT participants, 1134 women with a history of preeclampsia in one or more pregnancies were identified, and for 2/3 of the women, the diagnosis had been validated by scrutiny of hospital records[59]. A total of 2212 women with healthy, non-hypertensive pregnancies and a further 4000 women from the HUNT population provided the control group.

The HumanCoreExome-12 v1.0 (Illumina), HumanCoreExome-12 v1.1 (Illumina) and UM HUNT Biobank v1.0 was used to genotype ~600,000 variants. Genotype calling was performed with Genome Studio (Illumina). Samples with genotype call rates <99% were excluded, as were duplicates, samples with contamination >2.5% as estimated with BAF Regress, and those with gender mismatches or evidence of non-European ancestry from principal components analysis ($P < 0.0001$). Variants were excluded if the cluster separation score was <0.3, Gentrain score was >0.15, genotyping fail rate >1% or they deviated from Hardy–Weinberg equilibrium ($P < 1 \times 10^{-4}$). Samples were phased with Eagle2 v2.3[60], and genotype imputation was conducted with Minimac3 (v2.0.1, http://genome.sph.umich.edu/wiki/Minimac3)[61] and a merged reference panel that was constructed by combining the Haplotype Reference Consortium panel (release version 1.1)[62] and a local reference panel based on 2202 whole-genome sequenced HUNT study participants.

Association with preeclampsia was tested using a generalized mixed model including covariates birth year, sex, genotype batch, and principal components 1–4 as implemented in SAIGE. Principal components were computed using PLINK. Additional filters applied to the analysis included minor allele count ≥10 and imputation r2 ≥ 0.3. The study was approved by the Regional Committee for Medical and Health Research Ethics, Central Norway.

**UKBB data.** The UKBB project is a large prospective cohort study of ~500,000 individuals from across the United Kingdom, aged between 40 and 69 at recruitment[63]. Genotyping was performed using a custom-made Affymetrix chip, UK

BiLEVE Axiom87 in the first 50,000 participants, and with the Affymetrix UK Biobank Axiom array in the remaining participants. 95% of the signals were on both chips. Imputation was performed by Wellcome Trust Centre for Human Genetics using a combination of the HRC, 1000 Genomes phase 3 and the UK10K haplotype resources[64]. Association analysis was performed using software developed at deCODE genetics[37]. Information on UKBB traits used in the study is included in Supplementary Table 14. UK Biobank's scientific protocol and operational procedures were reviewed and approved by the North West Research Ethics Committee.

**deCODE data.** Information on secondary deCODE traits used is included in Supplementary Table 14.

**Meta-analysis.** Variants were matched between data sets on the basis of position (NCBI Genome Reference Consortium Build 37) and alleles. We included variants that were well-imputed (info > 0.8), with a minor allele frequency >0.5% and present in at least two data sets. This left 11,796,347 and 12,130,433 autosomal variants for the maternal and fetal analysis, respectively.

The meta-analyses were conducted using the fixed-effects inverse-variance method based on effect estimates and standard errors implemented in METAL[65]. Genomic control correction was applied prior to meta-analysis, where appropriate, using the genomic control option in METAL.

**Conditional analysis.** We applied approximate conditional analyses, implemented in the GCTA software[19], to the meta-analysis summary statistics to look for additional association signals at each of the genome-wide significant loci. We analyzed the European and Asian data sets separately, estimating LD between variants using sets of 8700 whole-genome sequenced Icelandic individuals for the European analysis and 1787 chip-typed Kazakh individuals for the Central Asian analysis. The European and Central Asian results were then meta-analyzed. The analysis was restricted to variants present in both the European and Asian data sets and within 1 Mb from the index variants. We tested 7098 variants on one locus in the fetal analysis and report two variants with conditional $P$-value < 7.0 ×10⁻⁶, and 11,844 variants at two loci in the maternal analysis with no variant reaching the conditional $P$-value < 4.2 ×10⁻⁶.

**Validation of imputation.** One variant, rs139106685 on 6q14 showed suggestive association in the Central Asian maternal meta-analysis ($P = 4.02 \times 10^{-8}$) (Supplementary Fig. 1). This variant was absent in the European data. The minor allele frequency of this variant is 0.028 in the Central Asian data and this prompted us to check the imputation quality of the variant by direct genotyping of 4221 Kazakh samples. Twelve out of 134 imputed carriers and three out of 4087 imputed non-carriers carried the variant. The Central Asian data were imputed based on 1000 genomes phase 3 reference set complemented by a reference set based on whole-genome sequencing of 200 Kazakhs and Uzbeks (Supplementary Note 1) and this variant was not present in the Central Asian reference set. We thus conclude that the association observed for this variant is an artifact of low-quality imputation.

**Power calculation.** We estimated OR for a given allelic frequency of a variant for which we have 80% power to detect association using the estimated effective number of cases and controls for the study. The effective number of cases, or controls, was estimated for each cohort as $n_{eff} = 2*n_a*n_c/(n_a + n_c)$, where $n_a$, $n_c$ are the number of cases and controls used. For the Icelandic cohort, the effective sample size was further divided by the estimated genomic inflation factor, $\lambda_g$, to adjust for relatedness of the cohort. The effective sample size for the study was then calculated as the sum of $n_{eff}$ for individual cohorts.

**Functional annotation of *FLT1* variants.** Variants in LD with the lead variants were identified based on whole-genome sequenced Icelandic individuals using $r^2 > 0.8$. These variants were then annotated by the intersection with chromatin immunoprecipitation (ChIP-seq) and DNase hypersensitivity (DHS) data derived from the ENCODE project (www.encodeproject.org). ChIP-seq data were downloaded in pre-processed (MACS v2 algorithm) bigWig format representing analysis of acetylation of lysine K27 of histone H3 (H3K27ac) and mono-methylation of lysine K4 of histone H3. The accession numbers used are listed out in Supplementary Data 1. The ChIP-seq signal $P$-values were adjusted by the Benjamini-Hochberg procedure to account for multiple hypotheses and thresholded at the 1% FDR significance level. DNase hypersensitivity data (DHS) were downloaded in a pre-processed format (Hotspot algorithm). We then made use of the Joint Effect of Multiple Enhancers (JEME) resource to find enhancer-gene targets[66].

**Preeclampsia onset and birth weight.** The association between fetal and maternal discovery variants and disease onset was tested in combined GOPEC, deCODE, MoBa, and SSI offspring and maternal data sets, respectively, where data on time of diagnosis was available (Supplementary Table 8). The combined analysis included 800 early-onset and 2660 late-onset preeclampsia offspring and up to 378,185 controls. The combined maternal analysis included 1797 early-onset and 3757 late-onset maternal cases and up to 144,355 controls. Early-onset was defined as

diagnosis before 34 weeks gestation except in the SSI samples where early-onset was defined as delivery before 34 weeks gestation. Association with birth weight was tested in data from the UKBB. These data were not adjusted for the gestational duration. We tested the effect of the fetal variant on self-reported own birth weight ($N = 236,507$) and the effect of the maternal variants on reported weight of the first child ($N = 178,241$). Birth weight data were inverse normal transformed and adjusted for year of birth and the first 10 principal components. For the 6 variants tested, the Bonferroni corrected level of significance is $P = 0.05/6 = 0.0083$.

**Concordance analysis**. All five of the genome-wide significant maternal pre-eclampsia variants were in strong LD with known variants associated with BP and the preeclampsia risk allele (allele with higher frequency in preeclampsia cases versus controls) was also the same allele known to be associated with higher BP at sentinel SNPs of all five known BP loci. Therefore, we hypothesized that some or all of the other 892 known BP variants listed in Supplementary Table 18 of Evangelou et al.[16] (the paper reports a total of 984 BP variants at 901 loci), may also exhibit a positive correlation of the known higher BP allele and preeclampsia (or gestational hypertension) risk allele identified in our meta-analyses. Under the null hypothesis that such positive correlation does not exist at any of the 892 BP variants, the probability of concordance of the preeclampsia (or gestational hypertension) risk allele and higher BP allele at each variant is random and equals 0.5; and since the 892 variants are not in LD and hence are independent, the null hypothesis probability of total concordances at the 892 variants can be calculated with a binomial distribution based on 892 binomial trials with 0.5 probability of trial success. To calculate the tail probabilities of the binomial distribution, we used the pbinom function implemented in R statistical software (www.r-project.org). The p-value of this two-tail binomial test is extremely low for each meta-analysis ($P < 2.9 \times 10^{-5}$ to $P < 1.1 \times 10^{-22}$) (Supplementary Table 6). Thus the null hypothesis can be rejected and we conclude that the higher BP allele and preeclampsia (or gestational hypertension) risk allele are positively correlated at some of the 892 known BP variants.

For these analyses, the high BP allele at each of the 892 variants was taken from the European results section of Supplementary Table 18 in Evangelou et al.[16] and the preeclampsia or gestational hypertension risk allele was the higher frequency allele in cases compared to controls at each variant as observed in the GWAS meta-analyses of European mothers, Central Asian mothers, or European-Central Asian mothers combined. In Supplementary Table 6, the European maternal preeclampsia analysis used all 892 SNPs since all the variants had been genotyped in some of the 6 European maternal data sets, the Central Asian PE mothers analysis used 883 of the 892 variants, one SNP was omitted due to its PE odds ratio being 1.0 (and thus it had no PE risk allele) while 8 other variants with missing data were omitted; the European and Central Asian PE mothers analysis used 891 of 892 variants and omitted one variant with PE odds ratio of 1.0; the European GH mothers analysis used 888 of the 892 variants with one SNP omitted due to its GH odds ratio being 1.0 while 3 other variants with missing data were omitted.

**Separation of fetal and maternal effects**. The family genotype data were jointly analyzed across the GOPEC and MoBa cohorts using a modified version of the EMIM method that allows for joint modeling across cohorts (Supplementary Note 2). EMIM models the multiplicative increase in disease risk relative to the homozygous reference genotype conferred by the presence of 1 or 2 risk alleles in the mother ($S_1$ or $S_2$) or the offspring ($R_1$ or $R_2$). We considered a multiplicative effects model $\mathcal{L}(R_1, S_1)$ where $R_2 = R_1^2$ and $S_2 = S_1^2$. We then used a likelihood ratio test to formally test the significance of the full multiplicative effects model $\mathcal{L}(R_1, S_1)$ vs the models where the association is explained purely by the fetal effect only; $\mathcal{L}(R_1, S_1 = 1)$ and by the maternal effect only $\mathcal{L}(R_1, S_1 = 1)$. Both the likelihood ratio tests and confidence intervals show (Supplementary Data 4) that we are unable to reject the maternal only model at the 5% significance level for any of the 5 maternal loci, but conversely, we are able to reject the fetal only model for 4 of the 5 maternal loci.

The genotyping of the MoBa cohort is as previously described[6]. For three variants, rs259983, rs1421085, and rs10774624, follow-up data for the MoBa samples were in silico data based on 1046 fetal cases, 1469 maternal cases and 961 controls assayed on the Illumina HumanCoreExome-12 v1.1 chip and imputed based on the 1000 Genomes Project Phase 3 reference panel. Imputed variants are hard called with a hard call threshold of 0.1.

**LD-score regression**. We estimated the genetic correlation between pairs of traits using the cross-trait LD-score regression method[24,25] in our meta-analysis and summary statistics from traits in the deCODE and UKBB data sets. The traits tested were: diastolic blood pressure, systolic blood pressure, hypertension, coronary artery disease, ischemic stroke, venous thromboembolism, chronic kidney disease, type 2 diabetes, body mass index, birth weight of the first child, asthma, and rheumatoid arthritis. To avoid potential bias due to differences in LD structure between Central Asian and European data this analysis included only the European maternal meta-analysis. The sample size of the European offspring analysis was not sufficient for this analysis. We used results for about 1.1 million variants (excluding the MHC region), well-imputed in both data sets, and for LD information we used pre-computed LD scores for European populations (downloaded from https://data.

broadinstitute.org/alkesgroup/LDSCORE/eur_w_ld_chr.tar.bz2). To avoid bias due to overlapping samples, we calculated the genetic correlation between a meta-analysis of GOPEC, ALSPAC, and MoBa preeclampsia cohorts and the Icelandic GWAS summary statistic for each secondary trait, and also between a meta-analysis of deCODE, SSI and FINRISK preeclampsia cohorts and UKBB GWAS summary statistic for each secondary trait. The results of the two analyses were subsequently meta-analyzed. The birth weight analysis included included all European non-UK preeclampsia data sets and UKBB birth weight data. The sample size for each secondary trait is reported in Supplementary Table 11.

We estimated observed scale SNP heritability with LD-score regression using pre-computed LD variants found for about 1.2 million variants found in European populations (downloaded from: https://data.broadinstitute.org/alkesgroup/LDSCORE/eur_w_ld_chr.tar.bz2) for an effective sample size of 10,255 maternal and 7259 fetal cases in the European meta-analysis.

**Heritability estimation**. For each cohort, we applied GCTA (version 1.93.2) separately to the maternal and fetal case–control post-QC and pre-imputation genotypes. The genetic relatedness matrix was calculated on autosomal SNPs with –grm-cutoff=0.05 and the first 10 principal components were included as fixed-effects in the linear mixed model. The disease prevalence was set to 4%. The per-region heritability was calculated by combining the per cohort estimated heritability and standard error using fixed-effect inverse-variance meta-analysis. No significant heterogeneity was observed ($P_{het} > 0.1$).

**Polygenic risk score analysis**. We used PRS analyses of the GWAS results for one trait to investigate its predictive power for another trait. We used effect estimates based on GWAS analysis in the UKBB data set for hypertension and birth weight, and published meta-analysis for T2D[67], BMI[68], and CAD[69]. To avoid confounding the association, the analysis was not done on the same populations as we are used to deriving effect estimates.

The risk scores were calculated using genotypes for about 600,000 well-imputed autosomal markers. We estimated LD between markers using 4000 phased Icelandic samples and used this LD information to calculate adjusted effect estimates using LDpred[70]. We created several PRSs assuming different fractions of causal markers (the P parameter in LDpred), and selected the PRSs that best predicted the trait itself. The fraction of causal markers used was: 30% for the PRSs for hypertension and birth weight, 3% for the PRS for BMI, and 1% for the PRSs for T2D and CAD. The correlation between the PRS and traits was calculated using logistic regression in R (v3.5) (http://www.R-project.org) adjusting for principal components by including them as covariates in the analysis.

When analyzing the association of PRS for different traits on deCODE sets of preeclampsia and gestational hypertension the controls were the same as included in the GWAS analysis (female subjects recruited through different genetic research projects at deCODE, see description of deCODE data set above). For comparison of hypertension risk scores between traits in Iceland, and between cohorts, for consistency we excluded from the deCODE control group any individuals that were cases in any of the three traits tested (hypertension, gestational hypertension, preeclampsia), using the same, essentially hypertension-free, female control group for all three traits. Similarly, we only used female hypertension cases in this analysis.

We estimated the variance explained, $r^2$, using the method of Nagelkerke[71]. The reported variance explained is the estimated $r^2$ value for the model including the PRS and covariates, minus the $r^2$ for the model only including covariates.

**Reporting summary**. Further information on research design is available in the Nature Research Reporting Summary linked to this article.

## Data availability

Meta-analyzed GWAS data used in this study as well as individual-level GWAS data from the Uzbek and Kazakh studies and whole-genome sequencing data on Uzbek and Kazakh subjects have been deposited in the European Genome-phenome Archive (https://ega-archive.eu) under accession numbers are as follows. Whole-Genome Sequencing: EGAD00001005467, EGAD00001005466; Kazakhstan GWAS Genotypes: EGAD00010001945, EGAD00010001949, EGAD00010001947; Uzbekistan GWAS Genotypes: EGAD00010001917, EGAD00010001918, EGAD00010001919; GWAS Meta-Analyses: EGAD00010001983, EGAD00010001984, EGAD00010001985, EGAD00010001986, EGAD00010001987, EGAD00010001988. For pre-computed LD scores for European populations, see https://data.broadinstitute.org/alkesgroup/LDSCORE/eur_w_ld_chr.tar.bz2; for ENCODE project see www.encodeproject.org.

## Code availability

All custom codes used in this study are freely available online.

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

## Acknowledgements

Research leading to these results was conducted as part of the InterPregGen study, which received funding from the European Union Seventh Framework Programme under grant agreement no. 282540 and was supported by Wellcome Trust grant 098051. Some data used for the research were obtained from THL Biobank. We thank all study participants for their generous participation at THL Biobank. Part of this work was conducted using the UK Biobank Resource under application number 24711. A full list of acknowledgments appears in Supplementary Note 3.

## Author contributions

Manuscript preparation: V.S., R.M., N.O.W., L.St., and L.M. All authors contributed to critical revision of the manuscript. Study design: V.S., R.M., N.O.W., G.T., A.F.D., J.P.C., F.D., T.A., T.H., M.M., D.N., F.G., L.C.V.T., F.B.P., N.K., Z.M., S.M.E., N.A.B.S., V.A.D., J.J.W., U.T., A.-C.I., B.F., D.A.L., H.A.B., P.M., H.L., N.Z., G.S., K.S., and L.M. Phenotyping: J.F., K.M.A., G.B., V.A.D., S.M.E., Y.F., R.T.G., F.G., S.G., Q.H., A.H., S.H., T.Ja., I.J., T.Ju., N.K., K.Ki., K.Kl., Z.M., A.M., D.N., F.N., T.O., M.P., F.B.P., L.P., S.Sa., D.S.,

L.Sk., L.C.V.T., V.T., L.T., N.A.B.S., The FINNPEC Consortium, The GOPEC Consortium, A.F.D., J.J.W., B.F., H.A.B., P.M., H.L., G.S., and L.M. Genotyping and quality control: V.S., R.M., N.O.W., L.St., G.T., S.Sh., G.B., M-C.B., S.B., J.B.-G., I.C., S.M.E., C.S.F., Q.H., D.M.H., T.H., H.J., A.K., W.K.L., P.J.S., K.M.S., L.C.V.T., B.F., P.M., N.Z., and G.S. Data analysis: V.S., R.M., N.O.W., L.St., G.T., J.F., J.K.S., M-C.B., I.C., F.D., C.S.F., M.L.F., F.G., D.F.G., T.Ja., J.P.K., K.Ki., G.P., L.Sk., O.A.S., E.S.-U., B.F., and H.L. Interpretation: V.S., R.M., N.O.W., G.T., J.P.C., M-C.B., T.A., Z.M., S.M.E., Q.H., N.A.B.S., U.T., A.-C.I., L.C.V.T., P.M., D.A.L., H.L., N.Z., G.S., K.S., and L.M.

## Competing interests

V.S., L.St., G.T., J.K.S., O.A.S., M.L.F., S.G., A.H., V.T., H.J., T.Ju., T.O., S.Sa., I.J., U.T., D.F.G. and K.S., are employees of deCODE genetics/Amgen Inc. and declare competing financial interest. All remaining authors declare no competing interests.

## Additional information

Valgerdur Steinthorsdottir[1,64✉], Ralph McGinnis[2,64✉], Nicholas O. Williams[2,64], Lilja Stefansdottir[1,64], Gudmar Thorleifsson[1], Scott Shooter[2], João Fadista[3,4], Jon K. Sigurdsson[1], Kirsi M. Auro[5], Galina Berezina[6], Maria-Carolina Borges[7,8], Suzannah Bumpstead[2], Jonas Bybjerg-Grauholm[9], Irina Colgiu[2], Vivien A. Dolby[10], Frank Dudbridge[11], Stephanie M. Engel[12], Christopher S. Franklin[2], Michael L. Frigge[1], Yr Frisbaek[13], Reynir T. Geirsson[13], Frank Geller[3], Solveig Gretarsdottir[1], Daniel F. Gudbjartsson[1,14], Quaker Harmon[15], David Michael Hougaard[9], Tatyana Hegay[16], Anna Helgadottir[1], Sigrun Hjartardottir[13], Tiina Jääskeläinen[17], Hrefna Johannsdottir[1], Ingileif Jonsdottir[1,18], Thorhildur Juliusdottir[1], Noor Kalsheker[19], Abdumadjit Kasimov[16], John P. Kemp[20,7], Katja Kivinen[21], Kari Klungsøyr[22,23], Wai K. Lee[24], Mads Melbye[3,25,26], Zosia Miedzybrodska[27], Ashley Moffett[28], Dilbar Najmutdinova[29], Firuza Nishanova[29], Thorunn Olafsdottir[1,18], Markus Perola[5,30], Fiona Broughton Pipkin[31], Lucilla Poston[32], Gordon Prescott[27,33], Saedis Saevarsdottir[1], Damilya Salimbayeva[6], Paula Juliet Scaife[31], Line Skotte[3], Eleonora Staines-Urias[11], Olafur A. Stefansson[1], Karina Meden Sørensen[34], Liv Cecilie Vestrheim Thomsen[35,36], Vinicius Tragante[1,37], Lill Trogstad[38], Nigel A. B. Simpson[39], FINNPEC

Consortium*, GOPEC Consortium*, Tamara Aripova[16], Juan P. Casas[40,41], Anna F. Dominiczak [24], James J. Walker [10], Unnur Thorsteinsdottir[1,18], Ann-Charlotte Iversen [36], Bjarke Feenstra [3], Deborah A. Lawlor [7,8,42], Heather Allison Boyd [3], Per Magnus[43], Hannele Laivuori [17,44,45], Nodira Zakhidova[16], Gulnara Svyatova [6], Kari Stefansson [1,18] & Linda Morgan [19]

[1]deCODE genetics/Amgen Inc., Reykjavik, Iceland. [2]Wellcome Sanger Institute, Cambridge, UK. [3]Department of Epidemiology Research, Statens Serum Institut, Copenhagen, Denmark. [4]Department of Clinical Sciences, Lund University Diabetes Centre, Malmö, Sweden. [5]Finnish Institute for Health and Welfare, Helsinki, Finland. [6]Scientific Center of Obstetrics, Gynecology and Perinatology, Almaty, Kazakhstan. [7]MRC Integrative Epidemiology Unit, University of Bristol, Bristol, UK. [8]Population Health Science, Bristol Medical School, University of Bristol, Bristol, UK. [9]Department for Congenital Disorders, Danish Centre for Neonatal Screening, Statens Serum Institut, Copenhagen, Denmark. [10]Leeds Institute of Medical Research (LIMR), School of Medicine, University of Leeds, Leeds, UK. [11]Department of Non-Communicable Disease Epidemiology, London School of Hygiene and Tropical Medicine, London, UK. [12]Department of Epidemiology, Gillings School of Global Public Health, University of North Carolina at Chapel Hill, Chapel Hill, NC, USA. [13]Department of Obstetrics and Gynecology, Landspitali University Hospital, Reykjavik, Iceland. [14]School of Engineering and Natural Sciences, University of Iceland, Reykjavik, Iceland. [15]Epidemiology Branch, National Institute of Environmental Health Sciences, Durham, NC, USA. [16]Institute of immunology and human genomics, Uzbek Academy of Sciences, Tashkent, Uzbekistan. [17]Medical and Clinical Genetics, University of Helsinki and Helsinki University Hospital, Helsinki, Finland. [18]Faculty of Medicine, University of Iceland, Reykjavik, Iceland. [19]School of Life Sciences, University of Nottingham, Nottingham, UK. [20]The University of Queensland Diamantina Institute, The University of Queensland, Woolloongabba, QLD, Australia. [21]Division of Cardiovascular Medicine, University of Cambridge, Cambridge, UK. [22]Division of Mental and Physical Health, Norwegian Institute of Public Health, Oslo, Norway. [23]Department of Global Public Health and Primary Care, University of Bergen, Bergen, Norway. [24]Institute of Cardiovascular and Medical Sciences, BHF Glasgow Cardiovascular Research Centre, University of Glasgow, Glasgow, UK. [25]Department of Clinical Medicine, University of Copenhagen, Copenhagen, Denmark. [26]Department of Medicine, Stanford University School of Medicine, Stanford, CA, USA. [27]Division of Applied Medicine, School of Medicine, Medical Sciences, Nutrition and Dentistry, University of Aberdeen, Aberdeen, UK. [28]Department of Pathology, University of Cambridge, Cambridge, UK. [29]Republic Specialized Scientific Practical Medical Centre of Obstetrics and Gynecology, Tashkent, Uzbekistan. [30]Research Program for Clinical and Molecular Metabolism, Faculty of Medicine, University of Helsinki, Helsinki, Finland. [31]School of Medicine, University of Nottingham, Nottingham, UK. [32]Department of Women and Children's Health, King's College London, London, UK. [33]Lancashire Clinical Trials Unit, University of Central Lancashire, Preston, UK. [34]The Danish National Biobank, Statens Serum Institut, Copenhagen, Danmark. [35]Department of Clinical Science, Centre for Cancer Biomarkers CCBIO, University of Bergen, Bergen, Norway. [36]Department of Clinical and Molecular Medicine, Centre of Molecular Inflammation Research (CEMIR), Norwegian University of Science and Technology (NTNU), Trondheim, Norway. [37]Division Heart & Lungs, Department of Cardiology, University Medical Center Utrecht, University of Utrecht, Utrecht, The Netherlands. [38]Department of Infectious Disease Epidemiology and Modelling, Norwegian Institute of Public Health, Oslo, Norway. [39]Division of Womens and Children's Health, School of Medicine, University of Leeds, Leeds, UK. [40]Massachusetts Veterans Epidemiology Research and Information Center (MAVERIC), VA Boston Healthcare System, Boston, MA, USA. [41]Department of Medicine, Brigham and Women's Hospital, Harvard Medical School, Boston, MA, USA. [42]Bristol NIHR Biomedical Research Centre, Bristol, UK. [43]Centre for Fertility and Health, Norwegian Institute of Public Health, Oslo, Norway. [44]Institute for Molecular Medicine Finland, Helsinki Institute of Life Science, University of Helsinki, Helsinki, Finland. [45]Department of Obstetrics and Gynecology, Tampere University Hospital and Tampere University, Faculty of Medicine and Health Technology, Tampere, Finland. [64]These authors contributed equally: Valgerdur Steinthorsdottir, Ralph McGinnis, Nicholas O. Williams, Lilja Stefansdottir. *Lists of authors and their affiliations appears at the end of the paper. ✉email: valgerdur.steinthorsdottir@decode.is; rm2@sanger.ac.uk

## FINNPEC Consortium

Hannele Laivuori[17,44,45], Seppo Heinonen[46], Eero Kajantie[47,48,49,50], Juha Kere[51,52,53], Katja Kivinen[21] & Anneli Pouta[54]

[46]Obstetrics and Gynecology, University of Helsinki and Helsinki University Hospital, Helsinki, Finland. [47]PEDEGO Research Unit, Medical Research Center Oulu, Oulu University Hospital and University of Oulu, Oulu, Finland. [48]Public Health Promotion Unit, National Institute for Health and Welfare, Helsinki and Oulu, Finland. [49]Children's hospital, University of Helsinki and Helsinki University Hospital, Helsinki, Finland. [50]Department of Clinical and Molecular Medicine, Norwegian University of Health and Technology, Trondheim, Norway. [51]Department of Biosciences and Nutrition, Karolinska Institutet, Huddinge, Sweden. [52]Folkhälsan Institute of Genetics and Molecular Neurology Research Program, University of Helsinki, Helsinki, Finland. [53]School of Basic & Medical Biosciences, King's College London, London, UK. [54]Department of Government Services, National Institute for Health and Welfare, Helsinki, Finland.

## GOPEC Consortium

Linda Morgan[19], Fiona Broughton Pipkin[31], Noor Kalsheker[19], James J. Walker[10], Sheila Macphail[55], Mark Kilby[56], Marwan Habiba[57], Catherine Williamson[58], Kevin O'Shaughnessy[59], Shaughn O'Brien[60], Alan Cameron[24], Christopher W. G. Redman[61], Martin Farrall[62], Mark Caulfield[63] & Anna F. Dominiczak[24]

[55]Newcastle upon Tyne Hospitals NHS Foundation Trust, Newcastle upon Tyne, UK. [56]The Centre for Women's & Newborn Health, College of Medical and Dental Sciences, University of Birmingham, Birmingham, UK. [57]University of Leicester, Leicester, UK. [58]Division of Women's Health, Kings College London, London, UK. [59]Department of Medicine, University of Cambridge, Cambridge, UK. [60]Keele University School of Medicine, Stoke-on-Trent, UK. [61]Nuffield Department of Obstetrics and Gynaecology, University of Oxford, Oxford, UK. [62]Radcliffe Department of Medicine, University of Oxford, Oxford, UK. [63]William Harvey Research Institute, Barts and the London School of Medicine and Dentistry, London, UK.

