## [Peer Review File · Nature Communications]

Reviewers' comments:

Reviewer #1 (Remarks to the Author):

In this study Steinthorsdottir and collages performed a genome-wide association meta-analysis of preeclampsia investigating the effect of variants in both the maternal and fetal genome. The study include mostly European preeclamptic mothers as well as offspring from preeclamptic mothers. By folding in maternal and offspring samples from Central Asia, sample size for the maternal GWAS reached 9515 cases and around 160,000 controls, while sample size for the offspring GWAS is 6775 cases and around 400,000 controls. The authors confirm association with the fetal locus FLT1, and identify two novel maternal loci for preeclampsia (ZNF831 and FTO), which are known loci for blood pressure. By analysing only well-established loci for blood pressure, additionally three loci are identified. The authors perform advanced statistical analyses to fine map and understand the signals identified. Using polygenic risk score analyses, the authors demonstrate that preeclampsia is associated with an increased polygenic risk score for hypertension.

The major claims in the paper are in my opinion well justified. The data is technically sound, the paper provides strong evidence for its conclusions. The results are clearly presented, which is a major effort for such a statistically rich and dense paper, combining so many datasets. No additional analyses are need. All analyses are sufficiently described.

Perhaps a bit disappointing that no additional loci in the fetal genome was identified. There is a very useful power calculation included in the paper, suggesting that sample size is still too small to detect effect of low MAF variants.

A major strength of the paper is that the authors have a small sample of family genotype data allowing them to separate the fetal and maternal effect which is biologically interesting.

The paper will be of great interest to the field. It is the first to report loci for preeclampsia in the maternal genome. It will be influential because it is a role model paper for how to disentangles the effect of maternal and fetal genome on traits which are determined by both genomes.

Some minor comments are:

How fair is it to go down to minor allele frequency 0.005 (page 27) given the many different genotyping arrays used in the study?

How sure are the authors that the lead SNPs are pointing to the genes indicated?

Regarding Table 3. It is interesting that the lead variant in the FLT1 locus is so clearly associated only with late onset preeclampsia. Is it possible to look up in existing RNAseq datasets of placenta how the transcriptional pattern of FLT1 change during gestation? If possible, it would add a nice touch to the reader interested in the biology behind preeclampsia.

It would be useful for the scientific community if the meta-analyses are made publicly available after publication. Will the sumstat be available?

Reviewer #2 (Remarks to the Author):

Summary:

I commend the authors on such a comprehensive manuscript, containing lots of large, thorough analyses, which is clearly the product of a great deal of work by the whole team.

The two primary GWAS meta-analyses of preeclampsia include: (i) the first large-scale analysis of maternal preeclampsia which reports two novel genome-wide significant loci at ZNF831 and FTO loci and further associated variants at BP loci and (ii) an enlarged analysis of fetal variants which confirms the previously reported FLT1 locus and identifies a new independent signal at this locus. So it is particularly the analysis of maternal preeclampsia which shows the most progress here.

The many secondary analyses include: conditional analyses; lookups of BP variants and

concordance testing; comparisons of maternal and fetal effects; stratified analyses by onset of preeclampsia; association with birth weight; distinguishing between preeclampsia and gestational hypertension; genetic correlation analyses and PRS analyses to investigate overlapping associations with other traits. It is of note that several new GWAS analyses of other traits were performed by this team, in order to complete these secondary analyses, in addition to the primary preeclampsia meta-analyses.

I particularly commend the authors on the extra work of generating a novel sequencing reference panel of individuals from Kazakhstan and Uzbekistan to allow for analysis of central Asian subjects. The authors are also commended for the use of advanced statistical methodology, which goes beyond the standard methodological content routinely seen in GWAS papers, e.g. for concordance testing; EMIM testing of maternal & fetal effects; adjusted PRS analyses to test for independence.

Major Comments:

- 1) The results of the FTO variant from the maternal analysis in Table 1 suggest that this variant was not actually replicated. In the follow-up data alone the p-value 0.089 was non-significant. So the significance in the combined meta-analysis (described as "remained significant") is clearly driven by the high significance from the discovery analysis. It is therefore not clear from a lack of pre-specified significance reporting criteria, as to whether this locus should in fact be reported here and claimed as a novel locus, with only evidence from the discovery stage.
- 2) Following on from this, I can see that het p-values are presented in the sup tables to show no evidence of heterogeneity between the European and Central Asian samples within the discovery analyses. However, I cannot find any het p-values to test for heterogeneity between the discovery and replication samples. This would also help to confirm if there is support from the follow-up data for the signals being claimed.
- 3) The study design for the follow-up of the offspring analysis seems rather ad-hoc, with separate stages of follow-up of 4 variants in Kazakh samples and then follow-up of only the FLT1 variant in the Finnish samples as well as the Kazakh samples. Why weren't the Finnish samples also used to follow-up the other 3 variants?
- 4) I do not follow all the evidence completely, to fully believe or understand the final conclusion made from the concordance testing, that the high concordance indicates evidence of "true-positive preeclampsia susceptibility loci" (lines 198-201). Whilst I follow all of the statistical methodology explanation in the Methods (lines 664-717) for the concordance testing in general, and agree with the high concordance between BP and preeclampsia associations, I do not follow the concluding claim for being true preeclampsia loci themselves. Is there a supporting reference for this method and theory?
- 5) In the Discussion lines 340-342 suggest that preeclampsia is a polygenic trait. But there seems to be no mention of the heritability of preeclampsia in the article. Has this been estimated in previous studies? Could the authors also use their data to calculate the heritability of preeclampsia?
- 6) Furthermore, what proportion of variance do the identified loci explain for preeclampsia? There only seem to be percentage variance explained calculation for the hypertension PRS analyses instead.
- 7) Please provide methods text to describe the actual statistical model used for the preeclampsia GWAS analyses, e.g. which covariates were included in the models, etc. Did all cohorts follow the same analysis model? Even if the exact analysis model were cohort specific, it would help to have some general methods text overall, as well as a detailed description within each cohort study description in the Methods section. At the moment, most study descriptions focus on e.g. the case definition, the genotyping and QC, but do not describe the cohort-level analysis.

Minor Comments:

- 8) In addition to the cohort-specific descriptions of the control subjects used within the Methods text, I think the main section of the paper could benefit from a brief, general description of the eligibility criteria for the selected controls...especially when later analyses of gestational hypertension were more restrictive on the hypertension statuses of control subjects. For example, were the preeclampsia analysis controls checked to be free of gestational hypertension?
- 9) Despite the power calculations that were performed for the n-effective numbers of cases & controls and also the heterogeneity testing between Europeans and Central Asians within the meta-analyses...I notice that the ratio of cases to controls is very different between Europeans and Central Asians – could the authors comment to justify that this could not cause any imbalance or

bias within the meta-analysis?

10) Is there a supporting reference that can be cited to show whether the power calculation method described in Methods (lines 636-641) is novel here or standardly used elsewhere?

11) Please state how you have defined a locus within this paper.

12) From Evangelou et al, 2018, there are more than 896 BP variants reported within the total 901 loci. Please describe how only these 896 were selected or filtered for this study?

13) Later on in the Methods text (lines 714-717) it seems that some of the BP variants were omitted if they were unavailable in the GWAS results. However in the "preeclampsia and BP variants" results section it gives the impression that all 896 variants were able to be evaluated in the GWAS results. Please therefore state more clearly how many of the BP variants were covered, and how many variants required proxies, and what criteria were used for choosing proxies in LD. And then in line 714 please give the exact number, rather than the vague "a few of the 892 variants..."

14) For the "Preeclampsia and BP variants" section, it would help to clarify that these are just single-SNP lookups at this stage, i.e. to distinguish from the later section on PRS analyses.

15) Did the lookups only consider the exact BP variant itself, or did searches also include for example all other SNPs in high LD with them? I would recommend the latter, if this had not been done. Then extra clarity would help, especially as this paragraph seems overly long and complicated to then compare the LD of the variants from the preeclampsia follow-up with the BP-associated variants, when actually after the entire lookup of all 896 BP variants, only the same loci initially observed are the ones that reach significance, and no further ones are identified.

16) For the analyses corresponding to Sup Fig 5: Please emphasize more clearly that these BP effect estimates have come from completely new GWAS analyses that you have performed in deCODE and UKBB, i.e. even though the SNPs were selected from the Evangelou et al paper, that the GWAS results were not taken from their publicly available summary statistics. Why were the BP traits standardised with inverse normal transformation (as stated in Sup Table 17)? The x-axes of the plots would be more meaningful if they were in mmHg units. This would also enable the authors to clarify whether their beta estimates were similar to those reported by Evangelou et al.

17) For the "preeclampsia stratified by onset" section in the Results, it would help to briefly state the N sample sizes of the two early vs late onset subgroups here, as well as the further details provided in the Methods. Furthermore, please also state which ancestries the samples were in these stratified analyses?

18) I note that the birth weight data from UKBB is self-reported, which could lead to bias from inaccuracies. Are there any other datasets which could be used as a sensitivity analysis to confirm support for these findings? I also note that the maternal analysis uses the birth weight of the first child. Is there any reasoning as to why the focus is only on the first child? Or is this simply the only data available in UKBB?

19) Please state the significance threshold that was used for claiming a significant association for the "Effect on birth weight" analyses?

20) Neither the Results Section nor the Methods section states the full list of traits which are tested for genetic correlation: the list is only found within Fig2 or ST14. Please also state which traits were tested but not correlated.

21) Please state the significance threshold used for claiming a genetic correlation result to be significant. I note from ST14 that the correlation with BMI was $p=5.9e-3$ but this is not stated as being correlated, whereas the result for the reported negative correlation with birth weight is only slightly higher at $p=2e-4$.

22) The PRS analyses are said "to explore further the correlation between preeclampsia" and other traits...Please therefore explain why BMI was additionally tested in the PRS analyses, even though it was not reported to show genetic correlation?

23) In addition to the Methods text describing how LDpred was used to construct the PRS, please also clarify if all variants were considered for inclusion in the PRS, or whether any further p-value threshold cut-offs were applied? For each trait PRS it would also be helpful to know e.g: how many variants in the final PRS used; what percentage of variance of the corresponding trait does the PRS explain?

24) For the genetic correlation analyses and for the analyses leading to Sup Fig 5, SBP and DBP traits were considered as well as hypertension. Also the 896 variants selected from Evangelou et al, came from GWAS of quantitative BP traits. Please therefore explain the choice of generating a PRS for hypertension itself, rather than for any of the quantitative BP traits. Does the hypertension PRS have more power than the PRS for SBP or DBP would have, in order to explain a greater

proportion of the BP/hypertension trait variance?

25) In Sup Table 17, the trait description for SBP from deCODE says "see above", to suggest that it is identical to DBP. But could we check that the medication adjustment was +15 rather than +10, for example, rather than being exactly the same?

26) Please provide more detailed methods text for the conditional analysis approach using GCTA. For example, was a full genome-wide conditional analysis performed, or only region specific at the loci of interest?

27) In Table 2, please state the LD between the pairs of variants at the same locus.

Reviewer #3 (Remarks to the Author):

This manuscript reports result of a meta-analysis of GWAS data from eight studies including 9,515 women with preeclampsia from Europe and Central Asia and expands on a previous meta-analysis of offspring from preeclampsia pregnancies. Preeclampsia is a relatively common serious complication of pregnancy. It affects both maternal and fetal health and is a major cause of maternal and perinatal mortality. Results provide strong support for genetic variants near *FTL1* associated with preeclampsia in offspring of such pregnancies. The paper also reports the first genetic variants associated with preeclampsia in the maternal genome for variants on chromosome 20q13 near *ZNF831* and chromosome 16q12 near the *FTO* locus. These are previously established variants for blood pressure (BP) and further analysis of BP variants identified additional variants on three chromosomes associated with preeclampsia through the maternal genome. The authors go on to show that a polygenic risk score for hypertension associates with preeclampsia.

The two variants showing genome-wide association in mothers with preeclampsia were previously associated with blood pressure. The authors evaluated 896 established blood pressure variants and identified three loci that were significant after adjusting for testing the 896 variants. The association remains suggestive, although evidence on concordance for the direction of effects for preeclampsia and blood pressure provides support for the results.

The manuscript is well written and provides valuable results on genetic risk factors for preeclampsia, the relationship to genetic risk factors for blood pressure and subsequent risk of future cardiovascular disease.

Step by step responses to reviewers' comments on the manuscript 'Genetic predisposition to hypertension is associated with preeclampsia in European and Central Asian women'

We thank the reviewers for helpful comments and suggestions.

Reviewers' comments are in bold

Reviewer #1 (Remarks to the Author):

In this study Steinhorsdottir and collages performed a genome-wide association meta-analysis of preeclampsia investigating the effect of variants in both the maternal and fetal genome. The study include mostly European preeclamptic mothers as well as offspring from preeclamptic mothers. By folding in maternal and offspring samples from Central Asia, sample size for the maternal GWAS reached 9515 cases and around 160,000 controls, while sample size for the offspring GWAS is 6775 cases and around 400,000 controls. The authors confirm association with the fetal locus FLT1, and identify two novel maternal loci for preeclampsia (ZNF831 and FTO), which are known loci for blood pressure. By analysing only well-established loci for blood pressure, additionally three loci are identified. The authors perform advanced statistical analyses to fine map and understand the signals identified. Using polygenic risk score analyses, the authors demonstrate that preeclampsia is associated with an increased polygenic risk score for hypertension.

The major claims in the paper are in my opinion well justified. The data is technically sound, the paper provides strong evidence for its conclusions. The results are clearly presented, which is a major effort for such a statistically rich and dense paper, combining so many datasets. No additional analyses are need. All analyses are sufficiently described.

Perhaps a bit disappointing that no additional loci in the fetal genome was identified. There is a very useful power calculation included in the paper, suggesting that sample size is still too small to detect effect of low MAF variants.

A major strength of the paper is that the authors have a small sample of family genotype data allowing them to separate the fetal and maternal effect which is biologically interesting.

The paper will be of great interest to the field. It is the first to report loci for preeclampsia in the maternal genome. It will be influential because it is a role model paper for how to disentangles the effect of maternal and fetal genome on traits which are determined by both genomes.

Some minor comments are:

How fair is it to go down to minor allele frequency 0.005 (page 27) given the many different genotyping arrays used in the study?

We decided to include variants with MAF > 0.5% and that are present in at least two data sets in the analysis. This was done as a compromise since two European data sets (deCODE and FINRISK) were imputed based on population specific whole genome sequencing data yielding accurate imputation well below this MAF while the remaining sets were imputed with panels yielding lower confidence in this MAF range. For the SSI and MoBa data, variants with MAF < 1% were excluded from the analysis.

How sure are the authors that the lead SNPs are pointing to the genes indicated?

The lead SNPs reported here are (or in strong LD with) variants that have been previously reported to associate with traits that are relevant to preeclampsia, i.e. blood pressure and BMI. They have all been reported based on large studies but in most cases the reported gene is the nearest gene rather than being based on specific evidence regarding causality. Our study is small in comparison and we have not made any attempt at identifying the causal gene at each locus but rather used the gene name that has been conventionally linked to the association signals we identified.

Regarding Table 3. It is interesting that the lead variant in the FLT1 locus is so clearly associated only with late onset preeclampsia. Is it possible to look up in existing RNAseq datasets of placenta how the transcriptional pattern of FLT1 change during gestation? If possible, it would add a nice touch to the reader interested in the biology behind preeclampsia.

For study of the transcriptional pattern of Flt-1 during gestation it is important to compare samples from pregnancies in absence of labor, since labor markedly affects the transcriptome (refs: PMID: 20554320; PMID: 17823277). There are unfortunately no existing RNAseq data sets comparing different gestational ages based on pregnancies delivering by CS with no labor. The corresponding available microarray studies comparing different gestational ages based on deliveries by CS (refs: PMID: 19050320; PMID: 17170095) have unfortunately not included Flt-1 in their transcript's lists. Also, no available RNAseq data sets on gestational changes in the transcriptional pattern of Flt-1 were found for human tissues in the databases ENA European Nucleotide Archive, NCBI GEO Profiles and ArrayExpress.

It would be useful for the scientific community if the meta-analyses are made publicly available after publication. Will the sumstat be available?

The summary statistics will be made available at the European Genome-phenome Archive after publication as indicated under Data availability.

Reviewer #2 (Remarks to the Author):

Summary:

I commend the authors on such a comprehensive manuscript, containing lots of large, thorough analyses, which is clearly the product of a great deal of work by the whole team.

The two primary GWAS meta-analyses of preeclampsia include: (i) the first large-scale analysis of maternal preeclampsia which reports two novel genome-wide significant loci at ZNF831 and FTO loci and further associated variants at BP loci and (ii) an enlarged analysis of fetal variants which confirms the previously reported FLT1 locus and identifies a new independent signal at this locus. So it is particularly the analysis of maternal preeclampsia which shows the most progress here.

The many secondary analyses include: conditional analyses; lookups of BP variants and concordance testing; comparisons of maternal and fetal effects; stratified analyses by onset of preeclampsia; association with birth weight; distinguishing between preeclampsia and gestational hypertension; genetic correlation analyses and PRS analyses to investigate overlapping associations with other traits.

It is of note that several new GWAS analyses of other traits were performed by this team, in order to complete these secondary analyses, in addition to the primary preeclampsia meta-analyses. I particularly commend the authors on the extra work of generating a novel sequencing reference panel of individuals from Kazakhstan and Uzbekistan to allow for analysis of central Asian subjects. The authors are also commended for the use of advanced statistical methodology, which goes beyond the standard methodological content routinely seen in GWAS papers, e.g. for concordance testing; EMIM testing of maternal & fetal effects; adjusted PRS analyses to test for independence.

Major Comments:

1) The results of the FTO variant from the maternal analysis in Table 1 suggest that this variant was not actually replicated. In the follow-up data alone the p-value 0.089 was non-significant. So the significance in the combined meta-analysis (described as “remained significant”) is clearly driven by the high significance from the discovery analysis. It is therefore not clear from a lack of pre-specified significance reporting criteria, as to whether this locus should in fact be reported here and claimed as a novel locus, with only evidence from the discovery stage.

We do not determine the significance based on the results of the follow-up samples. They are much fewer than the discovery samples and hence are not well powered for replication. Instead we combine the results for the discovery and follow-up data and evaluate significance adjusting for all the 12 million variants tested.

2) Following on from this, I can see that het p-values are presented in the sup tables to show no evidence of heterogeneity between the European and Central Asian samples within the discovery analyses. However, I cannot find any het p-values to test for heterogeneity between the discovery and replication samples. This would also help to confirm if there is support from the follow-up data for the signals being claimed.

We thank the reviewer for pointing out that this information was missing. Heterogeneity p -values for test of heterogeneity between discovery and follow-up samples have now been added to Supplementary Table 6. We find $P_{\text{het}} > 0.05$ for all variants where we claim association.

3) The study design for the follow-up of the offspring analysis seems rather ad-hoc, with separate stages of follow-up of 4 variants in Kazakh samples and then follow-up of only the FLT1 variant in the Finnish samples as well as the Kazakh samples. Why weren't the Finnish samples also used to follow-up the other 3 variants?

As cited in footnote a, of Supplementary Table 2, the results shown for rs4769612 are proxy results from rs4769613 for genotyping of the FINNPEC fetal case-control dataset. The r^2 value for rs4769612 and rs4769613 is 1.0 in the Finnish population. These FINNPEC results for rs4769613 were taken from our 2017 paper which identified rs4769613 as being associated with susceptibility to preeclampsia in fetal cases (Nature Genetics 49:1255-1260, 2017). The FINNPEC results shown in Supplementary Table 2 were originally part of the follow-up genotyping reported in that 2017 paper which had a main focus on rs4769613 and the *FLT1* locus. For our 2017 paper, these rs4769613 FINNPEC follow-up fetal results were combined with follow-up genotyping results from rs4769613 in fetal cases and controls from the Norwegian MoBa dataset, and were reported in combined form (FINNPEC+MoBa) in Supplementary Table 1 of the 2017 paper. Subsequent to publication of the 2017 paper, GWAS genotypes became available for the MoBa fetal case-control dataset and are included in the GWAS Discovery fetal results and datasets being reported in the current paper under review. However, after the 2017 publication and

expiration of the InterPregGen Consortium grant that funded the study, the FINNPEC fetal sample DNA could no longer be genotyped. For completeness, we wanted to include the FINNPEC fetal results for rs4769613/rs4769612 that we already had; but FINNPEC fetal results for the other 3 variants in Supplementary Table 2 are missing because we could not genotype the FINNPEC fetal samples for the other 3 variants.

4) I do not follow all the evidence completely, to fully believe or understand the final conclusion made from the concordance testing, that the high concordance indicates evidence of “true-positive preeclampsia susceptibility loci” (lines 198-201). Whilst I follow all of the statistical methodology explanation in the Methods (lines 664-717) for the concordance testing in general, and agree with the high concordance between BP and preeclampsia associations, I do not follow the concluding claim for being true preeclampsia loci themselves. Is there a supporting reference for this method and theory?

SNP allele concordance at a large number of disease-associated SNPs has been previously analysed in some other contexts (e.g. to demonstrate that the same risk allele marks susceptibility to the same disease in different ethnic groups, see Mahajan et al. Nature Genetics 46:234-244, 2014). But we are unaware of a prior reference citing methods using SNP allele concordance as we did: (a) to conclude that allele concordance between susceptibility alleles for one phenotype mark true-positive loci for a second phenotype and (b) to estimate a lower bound for number of marked susceptibility loci for the second phenotype. So we decided to omit the lower bound estimate from our paper and also to restate our conclusions more in terms of what is statistically implied by the allele concordance results, namely, that at some known BP loci, the higher BP allele is positively correlated with the preeclampsia (or gestational hypertension) risk allele.

We have therefore omitted each sentence in Results specifying a lower bound for marked preeclampsia and gestational hypertension loci; and we also edited text in the Allele Concordance Methods section so our hypotheses and conclusions are now stated in terms of positive correlation between the higher BP allele and preeclampsia (or gestational hypertension) risk allele rather than the marking of true-positive susceptibility loci for preeclampsia (or gestational hypertension).

5) In the Discussion lines 340-342 suggest that preeclampsia is a polygenic trait. But there seems to be no mention of the heritability of preeclampsia in the article. Has this been estimated in previous studies? Could the authors also use their data to calculate the heritability of preeclampsia?

We thank the reviewer for the suggestion. We have now estimated the SNP heritability of preeclampsia. The results have been added to the Results section p. 10 and in a new Supplementary Table 13:

‘We made use of the genetic data to study the heritability of preeclampsia. We applied GCTA Genomic Relatedness Restricted Maximum Likelihood (GREML) analysis to the chip genotypes of European and Central Asian subjects to estimate the SNP heritability of preeclampsia on the liability scale¹⁹ (Methods). For Europe we found the heritability for maternal preeclampsia to be 38.1% (95% CI: 29.3-46.8) and in fetal preeclampsia 21.3% (95% CI: 7.4-35.3); for Central Asia the heritability was found to be 54.4% (95% CI: 29.6-79.3) and 42.5% (95% CI 17.3-67.7) for maternal and fetal preeclampsia respectively (Supplementary Table 13). These results are consistent with those previously reported in European family-based studies⁵.’

A paragraph has also been added to the Methods p. 32:

Heritability Estimation. For each cohort we applied GCTA (version 1.93.2) separately to the maternal and fetal case control post-QC and pre-imputation genotypes. The genetic relatedness matrix was calculated on autosomal SNPs with $-grm-cutoff=0.05$ and the first 10 principal components were included as fixed effects in the linear mixed model. The disease prevalence was set to 4%. The per region heritability was calculated by combining the per cohort estimated heritability and standard error using fixed effect inverse variance meta-analysis. No significant heterogeneity was observed ($P_{het} > 0.1$).'

6) Furthermore, what proportion of variance do the identified loci explain for preeclampsia? There only seem to be percentage variance explained calculation for the hypertension PRS analyses instead.

We have now calculated the variance explained by the five maternal risk variants and a sentence has been added to the Results p. 13:

'The corresponding PRS based on the five variants discovered here to associate with preeclampsia through the maternal genome only explains 0.25% of the variance in preeclampsia ($P = 1.6 \times 10^{-8}$).'

7) Please provide methods text to describe the actual statistical model used for the preeclampsia GWAS analyses, e.g. which covariates were included in the models, etc. Did all cohorts follow the same analysis model? Even if the exact analysis model were cohort specific, it would help to have some general methods text overall, as well as a detailed description within each cohort study description in the Methods section. At the moment, most study descriptions focus on e.g. the case definition, the genotyping and QC, but do not describe the cohort-level analysis.

We have now added a new paragraph to the Methods with a general description of the association analysis (p. 24):

'Association analysis: Logistic regression was used to test for association between variants and disease, assuming an additive model, treating disease status as the response and expected genotype counts from imputation as covariates. For the deCODE cohort information on county of origin within Iceland were included as covariates to adjust for possible population stratification. This was done using software developed at deCODE genetics. For the SSI cohort association analysis was done using PLINK. GOPEC, ALSPAC, MoBa and Uzbek cohorts included the top five ancestry principal components as covariates (SNPTEST (v2.4.1)). For the FINRISK and Kazakh cohorts the top twenty (FINRISK) and ten (Kazakhstan) ancestry principal components were included as covariates (SNPTEST (v2.5)).'

Minor Comments:

8) In addition to the cohort-specific descriptions of the control subjects used within the Methods text, I think the main section of the paper could benefit from a brief, general description of the eligibility criteria for the selected controls...especially when later analyses of gestational hypertension were more restrictive on the hypertension statuses of control subjects. For example, were the preeclampsia analysis controls checked to be free of gestational hypertension?

We have now added a sentence to the results section (p. 6) to clarify this: 'The controls comprised women with healthy pregnancies, with the exception of the two largest sample sets that used unselected (GOPEC) or female only (deCODE) population controls (Supplementary Table 1; Methods).'

9) Despite the power calculations that were performed for the n-effective numbers of cases & controls and also the heterogeneity testing between Europeans and Central Asians within the meta-analyses...I notice that the ratio of cases to controls is very different between Europeans and Central Asians – could the authors comment to justify that this could not cause any imbalance or bias within the meta-analysis?

The number of deCODE controls is especially large since they include a large part of the population of Iceland, and there are also considerably more controls than cases in some of the other European datasets and that is primarily why the ratio of cases to controls is so different between Europeans and Central Asians as noted by the reviewer. However, using the large number of controls should not bias the results for individual GWAS studies as it only provides more accurate estimate of the allelic frequency in the control group and hence increases power. The inverse variance method we used of combining effect size estimates, in essence, weights effects by GWAS sample size through the use of corresponding standard errors (as implemented by METAL software). This meta-analysis method of combining effect size estimates is well recognized and not known to bias results when the ratio of cases to controls is unequal (e.g. see “Comprehensive Literature Review and Statistical Considerations for GWAS meta-analysis” Nucleic Acids Research 40:3777-3784). Of course the algorithm does weight contributions by sample size, but these differences in weighting are statistically valid and do not undermine the validity of results and therefore differences in ratio of cases to controls should not be regarded as producing bias.

10) Is there a supporting reference that can be cited to show whether the power calculation method described in Methods (lines 636-641) is novel here or standardly used elsewhere?

The estimation of the effective sample size is commonly used and is for example described in „Quality control and conduct of genome-wide association meta-analyses.“ Winkler TW, et.al. Nat Protoc. 2014 May;9(5):1192-212. The addition here is that to account for relatedness of the Icelandic cohort we reduce the effective sample size by dividing by the genomic inflation factor which, in the case of the Icelandic cohort, is directly linked to the overestimation of „independent“ individuals in the study due to their relatedness.

11) Please state how you have defined a locus within this paper.

For the purpose of conditional analysis, we defined a locus as within one Mb of a given index variant.

12) From Evangelou et al, 2018, there are more than 896 BP variants reported within the total 901 loci. Please describe how only these 896 were selected or filtered for this study?

There was some inconsistency within the Evangelou et al, 2018 paper regarding the exact number of variants reported. We used Supplementary Table 18: Full association results for all novel and previously published SNPs etc. as our source for ‘all novel and previously published SNPs’.

13) Later on in the Methods text (lines 714-717) it seems that some of the BP variants were omitted if they were unavailable in the GWAS results. However in the “preeclampsia and BP variants” results section it gives the impression that all 896 variants were able to be evaluated in the GWAS results. Please therefore state more clearly how many of the BP variants were covered, and how many variants

required proxies, and what criteria were used for choosing proxies in LD. And then in line 714 please give the exact number, rather than the vague “a few of the 892 variants...”

We have removed the vague text in Methods and replaced it with text that gives exact numbers; proxies were considered but not used since results for all of the 892 variants (896 minus the four preeclampsia associated loci reported here) were available in at least one of our PE GWAS meta-analyses.

14) For the “Preeclampsia and BP variants” section, it would help to clarify that these are just single-SNP lookups at this stage, i.e. to distinguish from the later section on PRS analyses.

We have now stated this more clearly and the sentence on p. 7 now reads: Therefore, we evaluated the preeclampsia association in our meta-analysis **for each of the** 896 established BP variants, as reported by Evangelou et al.¹⁷.

15) Did the lookups only consider the exact BP variant itself, or did searches also include for example all other SNPs in high LD with them? I would recommend the latter, if this had not been done. Then extra clarity would help, especially as this paragraph seems overly long and complicated to then compare the LD of the variants from the preeclampsia follow-up with the BP-associated variants, when actually after the entire lookup of all 896 BP variants, only the same loci initially observed are the ones that reach significance, and no further ones are identified.

It is not clear to us that including variants that are highly correlated with the index blood pressure variants in the initial test would improve clarity of the presentation. As those variants are highly correlated with the index BP variants, they should give results similar to what the index variants are giving, both for blood pressure and for preeclampsia. And for any variant found in this search we would still have to describe how it relates to the index blood pressure variants and how it relates to any signal found for preeclampsia that might have been found at these loci. Thus, we would prefer to keep the approach we have been using. We have, however, shortened the paragraph describing these data (p. 7-8).

16) For the analyses corresponding to Sup Fig 5: Please emphasize more clearly that these BP effect estimates have come from completely new GWAS analyses that you have performed in deCODE and UKBB, i.e. even though the SNPs were selected from the Evangelou et al paper, that the GWAS results were not taken from their publicly available summary statistics. Why were the BP traits standardised with inverse normal transformation (as stated in Sup Table

17)? The x-axes of the plots would be more meaningful if they were in mmHg units. This would also enable the authors to clarify whether their beta estimates were similar to those reported by Evangelou et al.

Since the Evangelou et al. paper only reported summary statistics for the most significant trait (SBP, DBP or PP) for each variant we needed to obtain effect estimates for all variants for each trait elsewhere. The source of this data is stated both in the results section (p. 11) and in the legend to Supplementary Figure 5. We have now updated Supplementary Figure 5 with the effect on BP presented in mmHg as suggested.

17) For the “preeclampsia stratified by onset” section in the Results, it would help to briefly state the N sample sizes of the two early vs late onset subgroups here, as well as the further details provided in the

Methods. Furthermore, please also state which ancestries the samples were in these stratified analyses?

We have now added this information to the 'Preeclampsia stratified by onset' section in the Results. The sentence on p. 9 now reads: We therefore tested the association of our lead maternal and fetal variants with preeclampsia stratified by gestation at diagnosis of disease **in four European datasets of 1,797 and 3,757 early and late onset maternal cases and 800 and 2,660 early and late onset offspring, respectively** (Supplementary Table 11).

18) I note that the birth weight data from UKBB is self-reported, which could lead to bias from inaccuracies. Are there any other datasets which could be used as a sensitivity analysis to confirm support for these findings? I also note that the maternal analysis uses the birth weight of the first child. Is there any reasoning as to why the focus is only on the first child? Or is this simply the only data available in UKBB?

A birth weight meta-analysis from the Early Growth Genetics Consortium (EGG) consortium was recently published. That study includes a combination of data from EGG in combination with an earlier (smaller) version of the UKBB data we are using. Those data can, therefore, not be used for comparison. However, the EGG study did a comparison of the UKBB data and data from the consortium that was not self-reported and concluded that there were no major differences. In the results section we mention that one of the variants we report (at SH2B3) has previously been reported to associate with birth weight, consistent with our results. Our conclusion is that it is reasonable to use these data. Regarding using data on first child only this is the only available birth weight data in UKBB.

19) Please state the significance threshold that was used for claiming a significant association for the "Effect on birth weight" analyses?

A simple Bonferroni correction would put the significance threshold at: $0.05/6 = 0.0083$

20) Neither the Results Section nor the Methods section states the full list of traits which are tested for genetic correlation: the list is only found within Fig2 or ST14. Please also state which traits were tested but not correlated.

We have now added the full list of traits tested in the Methods section (p. 31). We have further added the number of traits tested in the Results section p. 11: Using the cross-trait LD score regression method^{23,24} we estimated the genetic correlation between preeclampsia, based on the maternal meta-analysis, and a selection of **12** relevant traits in deCODE and UKBB data (see Methods).

21) Please state the significance threshold used for claiming a genetic correlation result to be significant. I note from ST14 that the correlation with BMI was $p=5.9e-3$ but this is not stated as being correlated, whereas the result for the reported negative correlation with birth weight is only slightly higher at $p=2e-4$.

The significance threshold used for claiming genetic correlation is shown in the legend for Figure 2, $0.05/12 = 0.0042$. This has now been made clearer and now reads: **Significance threshold: $P = 0.05/12 = 0.0042$.**

22) The PRS analyses are said “to explore further the correlation between preeclampsia” and other traits...Please therefore explain why BMI was additionally tested in the PRS analyses, even though it was not reported to show genetic correlation?

We tested 12 traits (including SBP, DBP and hypertension) for genetic correlation. Five traits showed no evidence of genetic correlation ($P > 0.1$). In addition, BMI did not reach our significance threshold ($P = 0.0059$; significance threshold $P = 0.0042$). Given that obesity (high BMI) is considered one of the risk factors for preeclampsia we thought it was important to try to understand any potential genetic relationship between BMI and preeclampsia and, therefore, also included it in the PRS analysis.

23) In addition to the Methods text describing how LDpred was used to construct the PRS, please also clarify if all variants were considered for inclusion in the PRS, or whether any further p-value threshold cut-offs were applied? For each trait PRS it would also be helpful to know e.g: how many variants in the final PRS used; what percentage of variance of the corresponding trait does the PRS explain?

LDpred uses all markers that are included when creating the PRS, in this case about 600,000 markers. However, it reweights the effect estimates of the markers based on both LD between correlated markers and the assumption of how large fraction of markers are assumed to be causal. The assumption of the fraction of causal markers has similar effect as putting a P value threshold on the marker included, i.e. although all markers are included those that have high P values have very low weight. We have included in the methods section (p. 33) the assumed fraction of causal markers for the PRS we used and how much it explained of the variance in the trait itself. **The fraction of causal markers used was: 30% for the PRSs for hypertension and birth weight, 3% for the PRS for BMI and 1% for the PRSs for T2D and CAD.'**

24) For the genetic correlation analyses and for the analyses leading to Sup Fig 5, SBP and DBP traits were considered as well as hypertension. Also the 896 variants selected from Evangelou et al, came from GWAS of quantitative BP traits. Please therefore explain the choice of generating a PRS for hypertension itself, rather than for any of the quantitative BP traits. Does the hypertension PRS have more power than the PRS for SBP or DBP would have, in order to explain a greater proportion of the BP/hypertension trait variance?

The choice of using hypertension to create the polygenic risk score is simply because we consider hypertension to be a more relevant trait for preeclampsia. But as can be seen from the genetic correlation analysis its correlation with preeclampsia is similar to that of diastolic blood pressure and slightly greater than that of systolic blood pressure.

25) In Sup Table 17, the trait description for SBP from deCODE says “see above”, to suggest that it is identical to DBP. But could we check that the medication adjustment was +15 rather than +10, for example, rather than being exactly the same?

We thank the reviewer for catching this sloppy short-cut. Indeed the SBP data in Iceland was adjusted by adding 15 mmHg to the measured values as was done for the UK Biobank dataset. We have corrected this in Supplementary Table 18 (previously Supplementary Table 17).

26) Please provide more detailed methods text for the conditional analysis approach using GCTA. For

example, was a full genome-wide conditional analysis performed, or only region specific at the loci of interest?

We have now added the following paragraph to the Methods (p. 28): 'The analysis was restricted to variants present in both the European and Asian datasets and within 1 Mb from the index variants. We tested 7,098 variants on one locus in the fetal analysis and report two variants with conditional P-value < 7.0×10^{-6} , and 11,844 variants at two loci in the maternal analysis with no variant reaching the conditional P-value < 4.2×10^{-6} .'

27) In Table 2, please state the LD between the pairs of variants at the same locus.

The LD between the pairs of variants is reported in Supplementary Table 8. This information has now been added to the footnote of Table 2.

Reviewer #3 (Remarks to the Author):

This manuscript reports result of a meta-analysis of GWAS data from eight studies including 9,515 women with preeclampsia from Europe and Central Asia and expands on a previous meta-analysis of offspring from preeclampsia pregnancies. Preeclampsia is a relatively common serious complication of pregnancy. It affects both maternal and fetal health and is a major cause of maternal and perinatal mortality. Results provide strong support for genetic variants near FTL1 associated with preeclampsia in offspring of such pregnancies. The paper also reports the first genetic variants associated with preeclampsia in the maternal genome for variants on chromosome 20q13 near ZNF831 and chromosome 16q12 near the FTO locus. These are previously established variants for blood pressure (BP) and further analysis of BP variants identified additional variants on three chromosomes associated with preeclampsia through the maternal genome. The authors go on to show that a polygenic risk score for hypertension associates with preeclampsia.

The two variants showing genome-wide association in mothers with preeclampsia were previously associated with blood pressure. The authors evaluated 896 established blood pressure variants and identified three loci that were significant after adjusting for testing the 896 variants. The association remains suggestive, although evidence on concordance for the direction of effects for preeclampsia and blood pressure provides support for the results.

The manuscript is well written and provides valuable results on genetic risk factors for preeclampsia, the relationship to genetic risk factors for blood pressure and subsequent risk of future cardiovascular disease.

REVIEWERS' COMMENTS:

Reviewer #1 (Remarks to the Author):

The authors have satisfactorily responded to all questions.

Reviewer #2 (Remarks to the Author):

The authors have responded well to all the reviewer comments, and present a good revised manuscript.

There have been some nice additions to the manuscript, e.g. GCTA heritability analysis and % variance explained analysis.

There are also now some appreciated changes, e.g. regarding the interpretation of the concordance testing.

I am now satisfied regarding major comment (1), because even though the FTO result did not reach significance in the follow-up meta-analysis data alone, the authors do now provide heterogeneity p-values between the discovery vs replication data from the combined meta-analysis, which shows that there was no heterogeneity in the effect estimates between the discovery and follow-up data. Furthermore, I note that the authors have been careful with the language used, and only said "remained significant", so nowhere in the manuscript is there specific language claiming to have "replicated" the association in the follow-up data.

My only remaining minor comments are as follows:

(a) Re major comment (3), I ask the authors to include a bit of this brief explanation of the FINNPEC data availability within the FINNPEC study description paragraph, to help explain to the reader why not all variants are available here. The current footnote in ST2 that the authors refer to, state the r^2 for the proxy variant, but doesn't explain the data availability situation, or that this data has simply come from the previous 2017 publication.

(b) Re comment (6): For clarity and distinction of terminology, I would recommend that the authors use the language "genetic risk score" and abbreviation "GRS" to correspond to the genetic risk score of the five variants used to calculate the % variance explained. Then, instead, "PRS" can be reserved for meaning "polygenic risk scores" which by definition are expected to include all genome-wide SNPs, like according to the LDpred approach they have used.

(c) Furthermore, re comment (6), please could the authors also include some brief methods text on the new % variance explained analysis.

(d) Re comment (12), I think the confusion here is that only 896 variants have been used from the Evangelou et al 2018 paper, whereas this BP-GWAS paper reports a total of 901 loci. So it is confusing that the number in your paper is fewer. From Evangelou et al 2018, the 901 total loci include: 325 variants at the 325 novel 2-stage loci (in Sup Table 2) + 210 variants at the 210 novel 1-stage loci (in Sup Table 3) + 357 variants at the previously published 274 loci (in Sup Table 4) + 92 previously reported variants from Hoffman et al 2017 replicated for the first time (in Sup Table 5) = a total of 984 variants at 901 loci. Note that Sup Table 18 from Evangelou et al, which the authors have used here, is a Sup Table corresponding to results from a secondary analysis comparing results across different ancestries. So it may be that not all variants were available in each individual other ancestry. Indeed, there is a footnote comment in Sup Table 18 to suggest this. Furthermore, it is already shown in Sup Table 4 of Evangelou et al, that not all of the 357 variants at the previously published 274 loci were covered within the UKB data, hence also why not all of these previously published variants would be contained within Sup Table 18 either. If possible, I therefore recommend the authors to take the total list of all 984 variants from the total 901 loci from Sup Tables 2, 3, 4 & 5 of the Evangelou et al 2018 paper, instead of Sup Table 18. Or at least, to state in this paper, that the 896 variants used came from Sup Table 18 of the Evangelou et al paper, and are a subset of the 896 variants from the total 901 BP-associated loci reported in the Evangelou et al paper. This will then help to clarify the correct numbers of loci reported across papers in the literature, to avoid further confusion in the literature in the future.

And help to align this paper correctly to the information in the Evangelou et al 2018 paper that the authors are citing, for good consistency.

(e) Re comment (19), please could the authors therefore now provide this Bonferroni significance level threshold in the Methods text, and also highlight in bold text, those results that are significant in the birth weight results column of Table 3, with a footnote stating the according Bonferroni significance threshold.

(f) Re comment (23): Thank you for the further description of your LDpred approach. Knowing that LDpred does include all markers in the model, I am just still slightly surprised that the total number of markers is only 600,000. The methods text states these are "well-imputed autosomal markers". Please could you therefore provide your criteria for "well-imputed", and state the imputation quality threshold that was presumably used. Maybe also a MAF criteria was used? Knowing the total number of genome-wide imputed markers that there would be from the GWAS, I am assuming that this imputation quality filtering must therefore have been quite strict...which is fine, but just helpful to be stated.

Step by step responses to comments from reviewer 2 on the manuscript 'Genetic predisposition to hypertension is associated with preeclampsia in European and Central Asian women'

Reviewer's comments are in bold

Reviewer #2 (Remarks to the Author):

The authors have responded well to all the reviewer comments, and present a good revised manuscript.

There have been some nice additions to the manuscript, e.g. GCTA heritability analysis and % variance explained analysis.

There are also now some appreciated changes, e.g. regarding the interpretation of the concordance testing.

I am now satisfied regarding major comment (1), because even though the FTO result did not reach significance in the follow-up meta-analysis data alone, the authors do now provide heterogeneity p-values between the discovery vs replication data from the combined meta-analysis, which shows that there was no heterogeneity in the effect estimates between the discovery and follow-up data. Furthermore, I note that the authors have been careful with the language used, and only said "remained significant", so nowhere in the manuscript is there specific language claiming to have "replicated" the association in the follow-up data.

My only remaining minor comments are as follows:

(a) Re major comment (3), I ask the authors to include a bit of this brief explanation of the FINNPEC data availability within the FINNPEC study description paragraph, to help explain to the reader why not all variants are available here. The current footnote in ST2 that the authors refer to, state the r^2 for the proxy variant, but doesn't explain the data availability situation, or that this data has simply come from the previous 2017 publication.

We have now added a sentence to the Methods section p.25, line 580: **Offspring genotype data included in this study was generated in our previous study.**

(b) Re comment (6): For clarity and distinction of terminology, I would recommend that the authors use the language "genetic risk score" and abbreviation "GRS" to correspond to the genetic risk score of the five variants used to calculate the % variance explained. Then, instead, "PRS" can be reserved for meaning "polygenic risk scores" which by definition are expected to include all genome-wide SNPs, like according to the LDpred approach they have used.

We have now changed PRS to **genetic risk score** (p. 13, line 312).

(c) Furthermore, re comment (6), please could the authors also include some brief methods text on the new % variance explained analysis.

We have now added to the methods section p. 33, line 782. 'We estimated the variance explained, r^2 , using the method of Nagelkerke (ref). The reported variance explained is the estimated r^2 value for the model including the PRS and covariates, minus the r^2 for the model only including covariates.'

(d) Re comment (12), I think the confusion here is that only 896 variants have been used from the Evangelou et al 2018 paper, whereas this BP-GWAS paper reports a total of 901 loci. So it is confusing that the number in your paper is fewer. From Evangelou et al 2018, the 901 total loci include: 325 variants at the 325 novel 2-stage loci (in Sup Table 2) + 210 variants at the 210 novel 1-stage loci (in Sup Table 3) + 357 variants at the previously published 274 loci (in Sup Table 4) + 92 previously reported variants from Hoffman et al 2017 replicated for the first time (in Sup Table 5) = a total of 984 variants at 901 loci. Note that Sup Table 18 from Evangelou et al, which the authors have used here, is a Sup Table corresponding to results from a secondary analysis comparing results across different ancestries. So it may be that not all variants were available in each individual other ancestry. Indeed, there is a footnote comment in Sup Table 18 to suggest this. Furthermore, it is already shown in Sup Table 4 of Evangelou et al, that not all of the 357 variants at the previously published 274 loci were covered within the UKB data, hence also why not all of these previously published variants would be contained within Sup Table 18 either. If possible, I therefore recommend the authors to take the total list of all 984 variants from the total 901 loci from Sup Tables 2, 3, 4 & 5 of the Evangelou et al 2018 paper, instead of Sup Table 18. Or at least, to state in this paper, that the 896 variants used came from Sup Table 18 of the Evangelou et al paper, and are a subset of the 896 variants from the total 901 BP-associated loci reported in the Evangelou et al paper. This will then help to clarify the correct numbers of loci reported across papers in the literature, to avoid further confusion in the literature in the future. And help to align this paper correctly to the information in the Evangelou et al 2018 paper that the authors are citing, for good consistency.

Based on the reviewer's suggestion we have now added 'the paper reports a total of 984 BP variants at 901 loci' to the Methods section p. 30, line 694.

(e) Re comment (19), please could the authors therefore now provide this Bonferroni significance level threshold in the Methods text, and also highlight in bold text, those results that are significant in the birth weight results column of Table 3, with a footnote stating the according Bonferroni significance threshold.

We have added **Significance threshold: $P = 0.05/6 = 0.0083$** to the footnote of Table 3 and in the Methods section p. 29, line 688 we have added 'For the 6 variants tested the Bonferroni corrected level of significance is $P = 0.05/6 = 0.0083$ '. We have not highlighted significant results in the table in bold as the editors specifically request that italics or bold font are not used to convey emphasis (both in the main text and display items).

(f) Re comment (23): Thank you for the further description of your LDpred approach. Knowing that LDpred does include all markers in the model, I am just still slightly surprised that the total number of markers is only 600,000. The methods text states these are "well-imputed autosomal markers". Please could you therefore provide your criteria for "well-imputed", and state the imputation quality threshold that was presumably used. Maybe also a MAF criteria was used? Knowing the total number

of genome-wide imputed markers that there would be from the GWAS, I am assuming that this imputation quality filtering must therefore have been quite strict...which is fine, but just helpful to be stated.

The 600,000 variants used in the PRS analysis were selected as the set of genotyped variants on the Illumina Omni chip used to genotype most of the Icelandic samples. This set does not include rare variants and the variants are usually well imputed in most datasets. We did not include more variants as although Ldpred does adjust for correlation between variants, including too many correlated variants can still lead to problems.